# MuonSSM: Orthogonalizing State Space Models for Sequence Modeling

**Thai-Khanh Nguyen** [* 1]  **Ngoc-Bich-Uyen Vo** [* 2]  **Thieu N. Vo** [† 3]  **Tan M. Nguyen** [† 4]  **Cuong Pham** [† 2]

## Abstract

State space models (SSMs) have emerged as efficient linear-time alternatives to attention for long-sequence modeling. However, existing SSMs often suffer from instability and memory degradation over extended horizons due to poorly conditioned first-order updates and unbalanced update geometry. We introduce MuonSSM, a general framework that stabilizes SSM training by explicitly conditioning the geometry of memory updates rather than the recurrent transition matrix. MuonSSM augments SSMs with a momentum-based pathway and a lightweight Newton–Schulz transformation on low-rank input injections, yielding bounded and spectrally conditioned updates while preserving parallel scan complexity. Theory shows that MuonSSM improves gradient propagation, mitigates spectral amplification, and enriches memory representations over long horizons. Extensive experiments across language, vision, and time-series benchmarks show consistent gains in accuracy, robustness, and long-context performance when integrated into diverse SSM backbones. These results establish geometric conditioning of updates as a principled pathway to stable, scalable sequence modeling.

## 1. Introduction

Modeling sequences efficiently and reliably remains a central challenge in modern machine learning. Transformer-based architectures have achieved strong empirical success (Vaswani et al., 2017), but their quadratic complexity in sequence length limits scalability as contexts grow (Tay et al., 2020). This has renewed interest in state space models (SSMs), which offer linear-time alternatives based on recurrent or scan-based dynamics. Starting from S4 (Gu et al., 2021; 2020) and followed by architectures H3 (Fu et al., 2022), S5 (Smith et al., 2022), and Mamba (Gu & Dao, 2024), modern SSMs combine strong performance with hardware-efficient selective scans, making them a promising replacement for attention-based models on long sequences.

Despite their efficiency, scan-friendly SSMs can degrade over long horizons and deep stacks: common input-dependent affine updates remain fundamentally first-order and can suffer from poor long-range signal propagation and optimization instability (Dao & Gu, 2024; Pascanu et al., 2013). Recent variants mitigate this mainly via gating/normalization or input-dependent scaling (Liu et al., 2024; Yang et al., 2024a;b), which helps control magnitudes but does not introduce temporal inertia nor explicitly condition the geometry of accumulated updates.

In this work, we propose MuonSSM[1], a framework for stabilizing state space memory by augmenting SSMs with explicit momentum in their update dynamics. Our core insight is that memory updates in SSMs can be viewed through an online learning lens (Behrouz et al., 2025b; 2024): just as momentum methods improve optimization by accumulating gradient directions over time, introducing temporal inertia into state updates can improve long-horizon information propagation (Nguyen et al., 2020; Ma et al., 2022). MuonSSM incorporates a lightweight momentum pathway to accumulate update directions across timesteps, together with a parallelizable normalization of input-dependent updates to maintain well-conditioned memory evolution. Importantly, these modifications preserve the affine structure required for efficient parallel scans and do not increase asymptotic computational complexity.

**Contributions.** Our contributions are threefold:

(i) We introduce MuonSSM, a family of SSMs that integrate momentum-based second-order dynamics and implicit spectral conditioning into scan-based recurrence;

---

[*]Equal contribution [†]Co-last authors [1]Faculty of Information Technology, Dainam University, Hanoi University of Science and Technology, Hanoi 10000, Vietnam [2]Faculty of Artificial Intelligence, Posts and Telecommunications Institute of Technology, Hanoi 10000, Vietnam [3]Department of Computer Science, University of Bath, Bath BA2 7AY, UK [4]Departments of Mathematics, National University of Singapore, Singapore 119076, Singapore. Correspondence to: Tan M. Nguyen <tanmn@nus.edu.sg>, Cuong Pham <cuongpv@ptit.edu.vn>.

*Proceedings of the 43rd International Conference on Machine Learning*, Seoul, South Korea. PMLR 306, 2026. Copyright 2026 by the author(s).

---

[1]The name MuonSSM reflects conceptual inspiration from the Muon optimizer (Jordan et al., 2024), which demonstrated the practical effectiveness of geometric conditioning for stabilizing optimization. Unlike Muon, which acts on parameter gradients, MuonSSM applies this principle directly within state space memory updates.

*Table 1.* Special cases of recent SSMs recovered from the general associative memory update in Eq. (1).

| Model | $\alpha_t$ | $\beta_t$ | $\eta$ | Update Rule |
|---|---|---|---|---|
| Mamba (Gu & Dao, 2024) | $\alpha_t$ | 1 | 0 | $\mathbf{S}_t = \alpha_t \mathbf{S}_{t-1} + \mathbf{v}_t \mathbf{k}_t^\top$ |
| DeltaNet (Yang et al., 2024b) | 1 | $\beta_t$ | 1 | $\mathbf{S}_t = \mathbf{S}_{t-1}(\mathbf{I} - \beta_t \mathbf{k}_t \mathbf{k}_t^\top) + \beta_t \mathbf{v}_t \mathbf{k}_t^\top$ |
| Gated DeltaNet (Yang et al., 2024a) | $\alpha_t$ | $\beta_t$ | 1 | $\mathbf{S}_t = \mathbf{S}_{t-1}\big(\alpha_t(\mathbf{I} - \beta_t \mathbf{k}_t \mathbf{k}_t^\top)\big) + \beta_t \mathbf{v}_t \mathbf{k}_t^\top$ |
| LongHorn (Liu et al., 2024) | 1 | $\frac{\beta_t}{1+\beta_t \mathbf{k}_t^\top \mathbf{k}_t}$ | 1 | $\mathbf{S}_t = \mathbf{S}_{t-1}\big(\mathbf{I} - \frac{\beta_t}{1+\beta_t \mathbf{k}_t^\top \mathbf{k}_t}\mathbf{k}_t \mathbf{k}_t^\top\big) + \frac{\beta_t}{1+\beta_t \mathbf{k}_t^\top \mathbf{k}_t}\mathbf{v}_t \mathbf{k}_t^\top$ |

(ii) We provide a theoretical analysis showing that MuonSSM-style updates preserve parallelizability while improving gradient propagation; and

(iii) We demonstrate empirically that MuonSSM variants consistently improve accuracy, robustness, and length generalization across a wide range of modalities and SSM backbones, including language, vision, and time-series tasks.

**Organization.** We introduce the MuonSSM architecture in Section 2. Theoretical analysis regarding parallel associative structure and gradient propagation follows in Section 3, with detailed proofs deferred to the appendix. Section 4 presents experimental evaluations across language, vision, and time-series benchmarks, with additional empirical analysis in Section 5. Finally, Section 6 reviews related work, and Section 7 concludes the paper.

## 2. Methodology: MuonSSM

### 2.1. Background: State Space Models as Associative Memory

Following recent works providing a unified view of SSMs as online associative memory mechanisms (Yang et al., 2024a; Behrouz et al., 2025b), we adopt the framework where at each timestep $t$, the model maintains a memory matrix $\mathbf{S}_t \in \mathbb{R}^{d \times m}$ and receives a key-value pair $(\mathbf{k}_t, \mathbf{v}_t) \in \mathbb{R}^m \times \mathbb{R}^d$. The memory state is updated as

$$\mathbf{S}_t = \mathbf{S}_{t-1}\big(\alpha_t(\mathbf{I}_m - \beta_t \eta \mathbf{k}_t \mathbf{k}_t^\top)\big) + \beta_t \mathbf{v}_t \mathbf{k}_t^\top, \quad (1)$$

where $\alpha_t \in (0, 1]$ controls memory retention, $\beta_t > 0$ determines the update magnitude, $\eta$ modulates recall correction, and $\mathbf{I}_m \in \mathbb{R}^{m \times m}$ is the identity matrix. This formulation shows that recent SSM variants (Mamba, DeltaNet, Gated DeltaNet, LongHorn) primarily differ in their choices of scalar gates $\alpha_t$, $\beta_t$, and $\eta$ (see Table 1).

### 2.2. Limitations of First-Order Updates

All updates derived from Eq. (1) remain inherently first-order. The change in memory is a rank-one modification:

$$\Delta \mathbf{S}_t \propto (\mathbf{v}_t - \alpha_t \eta \mathbf{S}_{t-1} \mathbf{k}_t)\mathbf{k}_t^\top, \quad (2)$$

which is structurally confined to the span of the current key $\mathbf{k}_t$. Over long sequences, repeated rank-one updates can

induce:

1. **Spectral anisotropy**: Singular values become highly non-uniform.

2. **Gradient degradation**: Vanishing gradients through $\prod_{n=T}^t \mathbf{D}_n$ where $\mathbf{D}_n = \alpha_n(\mathbf{I}_m - \beta_n \eta \mathbf{k}_n \mathbf{k}_n^\top)$.

3. **Memory interference**: New updates can overwrite previous information.

Current SSMs address these issues only indirectly through scalar gating, normalization, or transition-matrix parameterization, such as HiPPO-inspired and diagonal SSM parameterizations (Gu et al., 2020; 2022b;a). In contrast, MuonSSM explicitly conditions the geometry of input-dependent memory updates while also introducing an additional momentum pathway for gradient propagation.

### 2.3. MuonSSM: Orthogonalizing State Space Models

To mitigate these limitations while preserving parallel scan efficiency, we propose *MuonSSM*. The key idea is to augment the memory dynamics with two simple components: a momentum pathway that accumulates update directions across timesteps, and a lightweight Newton–Schulz (NS) normalization applied to each low-rank input injection. These components condition memory updates while retaining the affine structure required for efficient parallel scans. Theoretical justification is provided in Section 3.

#### 2.3.1. MOMENTUM-AUGMENTED DYNAMICS

MuonSSM maintains an auxiliary momentum matrix $M_t \in \mathbb{R}^{d \times m}$, updated as

$$M_t = \gamma M_{t-1} + \text{NS}\big(\tau \beta_t v_t k_t^\top\big), \quad (3)$$

$$S_t = S_{t-1}\big(\alpha_t(I_m - \beta_t \eta k_t k_t^\top)\big) + M_t, \quad (4)$$

where $\gamma \in (0, 1]$ is the momentum decay coefficient and $\tau > 0$ scales the normalized update. The operator $\text{NS}(\cdot)$ is defined below.

#### 2.3.2. SINGLE-ITERATION NEWTON–SCHULZ

Each input injection $X_t = \tau \beta_t \mathbf{v}_t \mathbf{k}_t^\top$ is a rank-1 matrix whose sole nonzero singular value $\sigma_t = \tau \beta_t \|\mathbf{v}_t\| \|\mathbf{k}_t\|$

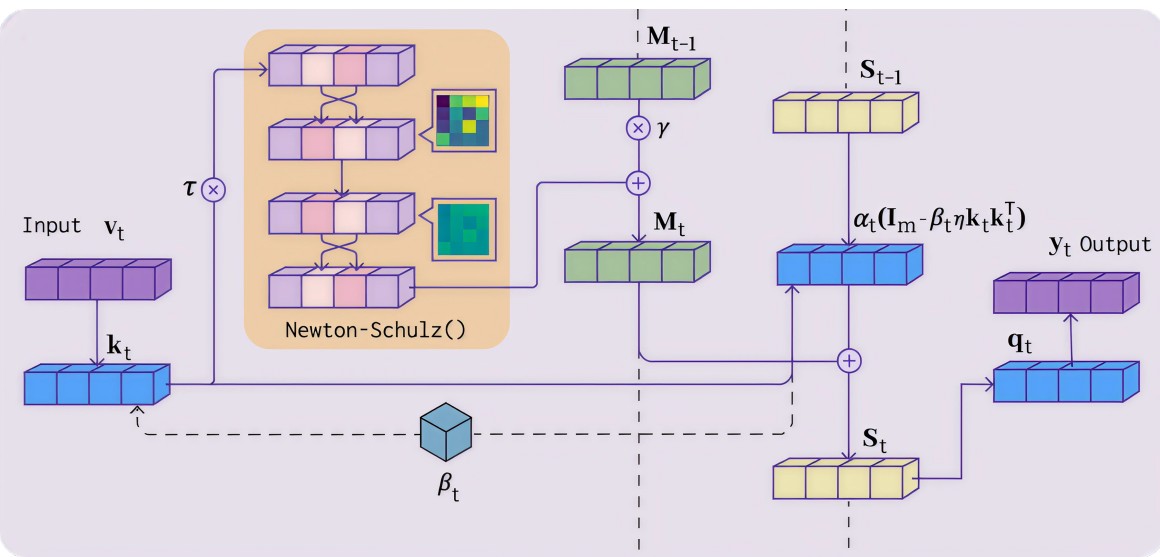

*Figure 1.* The MuonSSM architecture. At each timestep, the model forms a low-rank input injection $\tau \beta_t \mathbf{v}_t \mathbf{k}_t^\top$, applies a lightweight Newton–Schulz normalization, accumulates the normalized update through a momentum state $\mathbf{M}_t$, and updates the memory state $\mathbf{S}_t$ through an input-dependent transition. The output is computed as $\mathbf{y}_t = \mathbf{S}_t \mathbf{q}_t$.

varies arbitrarily across timesteps, producing spectrally unbalanced updates if left unnormalized. We apply a single-step Newton–Schulz iteration to each injection, which bounds singular values, preserves rank, and keeps per-step cost minimal without requiring an explicit SVD.

**Definition 2.1** (Single-iteration Newton–Schulz (NS)). For $X \in \mathbb{R}^{d \times m}$, define:

$$\text{NS}(X) = \left( a + b\, \tilde{X}\tilde{X}^\top + c \left( \tilde{X}\tilde{X}^\top \right)^2 \right) \tilde{X} \quad (5)$$

$$\text{where } \tilde{X} = \frac{X}{\max(\|X\|_F, \delta)} \quad (6)$$

Here $(a, b, c) = (3.4445, -4.7750, 2.0315)$ (Jordan et al., 2024), and $\delta > 0$ is a small constant to prevent division by zero.

The spectral properties of this operator and its backward-pass Jacobian geometry, which differs structurally from Frobenius normalization and drives rank enrichment in $\mathbf{M}_t$, are analyzed formally in Section 3.3.

## 3. Theoretical Analysis

We establish the key theoretical properties of MuonSSM: parallelizability, gradient stability, spectral conditioning, and rank enrichment. Together, these properties explain how MuonSSM mitigates the limitations of first-order SSMs while preserving computational efficiency.

### 3.1. Parallelizability

To enable parallel training, we cast the coupled memory and momentum updates (3)–(4) as a single block-affine recurrence, which admits efficient parallel associative scans.

**Proposition 3.1** (Block-Affine Recurrence). *Define the augmented state by horizontally concatenating the memory and momentum matrices:*

$$\mathcal{Z}_t = \begin{bmatrix} \mathbf{S}_t & \mathbf{M}_t \end{bmatrix} \in \mathbb{R}^{d \times 2m}.$$

*The coupled dynamics satisfy the linear recurrence:*

$$\mathcal{Z}_t = \mathcal{Z}_{t-1} \boldsymbol{\Phi}_t + \boldsymbol{\Psi}_t, \quad (7)$$

*where the transition matrix $\boldsymbol{\Phi}_t \in \mathbb{R}^{2m \times 2m}$ and input $\boldsymbol{\Psi}_t \in \mathbb{R}^{d \times 2m}$ are defined as:*

$$\boldsymbol{\Phi}_t = \begin{bmatrix} \mathbf{D}_t & \mathbf{0} \\ \gamma \mathbf{I}_m & \gamma \mathbf{I}_m \end{bmatrix}, \quad \boldsymbol{\Psi}_t = \begin{bmatrix} \mathbf{U}_t & \mathbf{U}_t \end{bmatrix} \quad (8)$$

*with $\mathbf{D}_t = \alpha_t(\mathbf{I}_m - \beta_t \eta \mathbf{k}_t \mathbf{k}_t^\top)$ and $\mathbf{U}_t = \text{NS}(\tau \beta_t \mathbf{v}_t \mathbf{k}_t^\top)$.*

*Remark* 3.2 (Parallel Training Efficiency). The block-affine recurrence (7) enables parallel associative scans (Blelloch, 1990) with operator $(\boldsymbol{\Phi}_i, \boldsymbol{\Psi}_i) \oplus (\boldsymbol{\Phi}_j, \boldsymbol{\Psi}_j) = (\boldsymbol{\Phi}_i \boldsymbol{\Phi}_j, \boldsymbol{\Psi}_i \boldsymbol{\Phi}_j + \boldsymbol{\Psi}_j)$. This reduces the recurrent depth from $O(L)$ to $O(\log L)$ while keeping total work linear in $L$. The NS pre-computation in Step 1 of Algorithm 1 is local to each timestep and adds only a constant-factor overhead.

Algorithm 1 summarizes the full parallel training procedure based on Proposition 3.1.

### 3.2. Gradient Stability

We next examine how the augmented dynamics affect gradient propagation through time. While standard SSMs suffer from repeated contraction induced by input-dependent transition matrices, the momentum pathway introduces an additional channel for gradient flow.

**Proposition 3.3** (Gradient Propagation in MuonSSM). *Let the augmented state $\mathcal{Z}_t = \begin{bmatrix} \mathbf{S}_t & \mathbf{M}_t \end{bmatrix}$ evolve according to the affine recurrence*

$$\mathcal{Z}_t = \mathcal{Z}_{t-1}\mathbf{\Phi}_t + \mathbf{\Psi}_t, \qquad \mathbf{\Phi}_t = \begin{bmatrix} \mathbf{D}_t & \mathbf{0} \\ \gamma\mathbf{I}_m & \gamma\mathbf{I}_m \end{bmatrix}.$$

*Then the gradient of the loss $\mathcal{L}$ with respect to $\mathcal{Z}_{t-1}$ propagates backwards via the transposed transition matrices:*

$$\frac{\partial \mathcal{L}}{\partial \mathcal{Z}_{t-1}} = \frac{\partial \mathcal{L}}{\partial \mathcal{Z}_T} \prod_{n=T}^{t} \mathbf{\Phi}_n^\top,$$

*which expands into the upper-triangular block form*

$$\frac{\partial \mathcal{L}}{\partial \mathcal{Z}_{t-1}} = \frac{\partial \mathcal{L}}{\partial \mathcal{Z}_T} \begin{bmatrix} \prod_{n=T}^{t} \mathbf{D}_n^\top & \sum_{k=t}^{T} \left(\prod_{j=T}^{k+1} \mathbf{D}_j^\top\right)(\gamma\mathbf{I}_m)^{k-t+1} \\ \mathbf{0} & (\gamma\mathbf{I}_m)^{T-t+1} \end{bmatrix}$$

*with the convention that any empty product equals $\mathbf{I}_m$.*

*Remark* 3.4 (Gradient Preservation via Scalar Momentum). When $\gamma \approx 1$, the momentum pathway $(\gamma\mathbf{I}_m)^{T-t+1}$ introduces a scalar eigenvalue close to unity into the Jacobian product $\prod_{n=T}^{t} \mathbf{\Phi}_n^\top$. Although the memory pathway $\prod_{n=T}^{t} \mathbf{D}_n^\top$ may contract due to input-dependent attenuation, a non-vanishing gradient component is preserved through the momentum state $\mathbf{M}_t$, mitigating exponential decay and enabling stable long-range credit assignment.

We emphasize that this analysis assumes a *linear recurrence*; additional nonlinearities and deep stacking in practice introduce interactions beyond the closed-form Jacobian of Proposition 3.3. Empirical support is provided in Appendix C and Figure 7, where gradient norm heatmaps confirm substantially more uniform long-range propagation.

### 3.3. Spectral Conditioning and Newton–Schulz Geometry

Beyond gradient flow, stable training requires controlling the spectral magnitude of memory updates. We analyze the NS operator in two complementary ways: its effect on singular values in the forward pass, and its distinct Jacobian geometry in the backward pass.

**Corollary 3.5** (Spectral Conditioning of Updates). *Let $\rho(\sigma) = a\sigma + b\sigma^3 + c\sigma^5$. Since $\|\tilde{\mathbf{X}}\|_F \leq 1$, all singular values of $\tilde{\mathbf{X}}$ lie in $[0,1]$, and the NS map sends each singular value $\sigma$ to $\rho(\sigma)$. Hence, for any input matrix $\mathbf{X}$,*

$$\sigma_{\max}(\mathrm{NS}(\mathbf{X})) \leq \sup_{\sigma \in [0,1]} |\rho(\sigma)| = 1 + \varepsilon_u. \qquad (9)$$

*For $(a,b,c) = (3.4445, -4.7750, 2.0315)$, we have $\rho(\sigma) \geq 0$ on $[0,1]$ and $\sup_{\sigma \in [0,1]} \rho(\sigma) \approx 1.2$. Moreover, since $\rho(\sigma) > 0$ for all $\sigma > 0$, NS preserves the rank of any nonzero input, and thus preserves the rank-one structure of each nonzero injection.*

Beyond the forward-pass bound of Corollary 3.5, the NS operator exerts a structurally distinct backward geometry that drives rank enrichment beyond what Frobenius normalization achieves.

**Proposition 3.6** (Backward Geometry of Newton–Schulz Normalization). *Let $\mathbf{X}_0 = \mathbf{u}\mathbf{w}^\top \in \mathbb{R}^{d \times m}$ be rank-one with $\|\mathbf{u}\|_2 = \|\mathbf{w}\|_2 = 1$. Denote by $\mathbf{u}_\perp$ and $\mathbf{w}_\perp$ any vectors orthogonal to $\mathbf{u}$ and $\mathbf{w}$ respectively, so that $\{\mathbf{u}\mathbf{w}^\top, \mathbf{u}_\perp\mathbf{w}^\top, \mathbf{u}\mathbf{w}_\perp^\top, \mathbf{u}_\perp\mathbf{w}_\perp^\top\}$ partitions $\mathbb{R}^{d \times m}$ into four orthogonal subspaces of dimensions $1$, $d-1$, $m-1$, and $(d-1)(m-1)$, respectively.*

*The Jacobian $D\mathcal{G}_{\mathbf{X}_0}$ of $\mathrm{NS}(X)$ (Definition 2.1) has the following eigenvalue families:*

$$\lambda = \begin{cases} 0 & \text{direction } \mathbf{u}\mathbf{w}^\top \\ a+b+c & \text{directions } \mathbf{u}_\perp\mathbf{w}^\top, \ \mathbf{u}\mathbf{w}_\perp^\top \\ a & \text{directions } \mathbf{u}_\perp\mathbf{w}_\perp^\top \end{cases} \qquad (10)$$

*The total dimension of each family is $1$, $d + m - 2$, and $(d-1)(m-1)$, respectively. For $(a,b,c) = (3.4445, -4.7750, 2.0315)$, these evaluate to $0, a+b+c \approx 0.701, a \approx 3.4445$.*

*In contrast, Frobenius normalization $\mathbf{X} \mapsto \mathbf{X}/\|\mathbf{X}\|_F$ has eigenvalue $0$ in the radial direction $\mathbf{u}\mathbf{w}^\top$ and eigenvalue $1$ in every other direction.*

*Remark* 3.7 (Implication for Rank Enrichment). Proposition 3.6 shows that NS and Frobenius normalization have different backward geometries. While both remove the radial direction $\mathbf{u}\mathbf{w}^\top$, Frobenius normalization leaves all tangent directions unchanged, whereas NS amplifies the fully orthogonal subspace $\mathbf{u}_\perp\mathbf{w}_\perp^\top$ by a factor $a \approx 3.4445$ and scales the mixed directions by $a + b + c \approx 0.701$. Thus, NS biases the backward signal toward directions orthogonal to the current rank-one write. When accumulated through the momentum recurrence, this provides a mechanism for producing less collinear updates and increasing the effective rank of $\mathbf{M}_t$, consistent with the ablation results in Appendix C.

### 3.4. Rank Enrichment

Finally, we analyze how momentum accumulation affects the representational capacity of the memory state. While each individual update is rank-1, their exponentially weighted accumulation leads to a progressively richer memory structure.

**Proposition 3.8** (Rank Enrichment via Momentum Accumulation). *Assume $\mathbf{M}_0 = \mathbf{0}$. The momentum state at time $t$ admits the expansion*

$$\mathbf{M}_t = \sum_{s=1}^{t} \gamma^{t-s} \mathrm{NS}\left(\tau\beta_s \mathbf{v}_s \mathbf{k}_s^\top\right). \qquad (11)$$

*Consequently,*

$$\mathrm{rank}(\mathbf{M}_t) \leq \min(t, d, m). \qquad (12)$$

*Moreover, this upper bound is attainable and generically tight: under non-degenerate updates ($\tau > 0$, $\beta_s > 0$, $\mathbf{v}_s \neq \mathbf{0}$, $\mathbf{k}_s \neq \mathbf{0}$), the set of direction pairs $\{(\mathbf{v}_s, \mathbf{k}_s)\}_{s=1}^t$ for which $\mathrm{rank}(\mathbf{M}_t) < \min(t, d, m)$ has measure zero with respect to the product surface measure on $(S^{d-1} \times S^{m-1})^t$.*

While momentum accumulation allows the memory to reach a rank up to $\min(t, d, m)$, rank alone is insufficient to characterize representational capacity. We therefore consider the *effective rank*

$$r_{\mathrm{eff}}(\mathbf{M}_t) = \frac{\left(\sum_i \sigma_i(\mathbf{M}_t)\right)^2}{\sum_i \sigma_i(\mathbf{M}_t)^2}, \qquad (13)$$

which measures how uniformly the singular values of $\mathbf{M}_t$ are distributed. A higher effective rank indicates that the momentum state uses more independent representational directions, reducing the risk that repeated rank-one writes collapse into a small subspace. Together with Proposition 3.6, this suggests that the backward geometry of NS can encourage less collinear future writes, while momentum accumulation integrates these writes over time. This mechanism is consistent with the three-way ablation in Appendix C, where replacing NS with Frobenius normalization reduces the effective rank and downstream accuracy.

We defer all proofs to Appendix A.

# 4. Experiments

We present a comprehensive empirical evaluation of MuonSSM across three distinct modalities: language, vision, and time series. Our primary objective is to assess whether the proposed memory-update mechanism translates into tangible performance gains and improved robustness across diverse data distributions. We benchmark MuonSSM against representative state-of-the-art SSM baselines, including Mamba, LongHorn, and Gated DeltaNet. To ensure a controlled comparison, we maintain identical parameter counts, architectural hyperparameters, and training budgets across all experiments, thereby isolating the effect of the MuonSSM update mechanism. All experiments are conducted on four NVIDIA H100 GPUs.

## 4.1. Language Modeling and Long-Context Retrieval

**Setup.** We investigate the capability of MuonSSM to handle discrete sequential data requiring both common-sense reasoning and precise long-range information retrieval. We adopt a 170M parameter configuration to conduct a controlled study on algorithmic efficiency in the small-scale regime. All models are pre-trained from scratch on the FineWeb-Edu 10B dataset (Lozhkov et al., 2024), a high-quality educational corpus chosen to evaluate the model's ability to acquire reasoning capabilities and linguistic structure efficiently under a controlled compute budget. Following pre-training, we perform supervised fine-tuning on the Alpaca-52K dataset (Taori et al., 2023) to unlock instruction-

following capabilities. Crucially, the SFT stage is conducted with a maximum sequence length of 2048 tokens. This constraint allows us to explicitly test the model's ability to extrapolate to longer contexts during evaluation, assessing whether the spectral properties of MuonSSM facilitate length generalization beyond the training horizon.

**Results.** As summarized in Tables 2 and 3, MuonSSM demonstrates superior performance compared to standard SSMs. On common-sense reasoning benchmarks derived from the FineWeb-Edu pre-training, the model achieves lower perplexity, suggesting that spectrally balanced updates facilitate better knowledge compression. More importantly, in the Single Needle in Haystack (S-NIAH) evaluation, MuonSSM maintains high retrieval accuracy across PassKey (PK), Number (N), and UUID tasks up to context lengths of 8K tokens. This result is particularly significant given that the instruction tuning was limited to 2048 tokens. Unlike baseline models which typically exhibit rapid performance degradation when extrapolating beyond their training context window, MuonSSM mitigates long-range gradient attenuation by introducing a momentum pathway and conditioning the geometry of input-dependent memory updates, thereby improving length generalization.

## 4.2. Vision Spatial Modeling and Robustness

**Setup.** Following the MambaVision framework (Hatamizadeh & Kautz, 2025), we evaluate MuonSSM as a drop-in replacement for visual state-space modeling. The architecture employs a hierarchical design where images are processed by a convolutional stem followed by stacked SSM mixers. We replace the standard Mamba mixers with MuonSSM blocks while keeping the hybrid attention layers and patch embedding strategies invariant. We utilize *Tiny* model variants to strictly control the computational budget. We assess performance on three standard tiers of visual understanding: image classification on ImageNet-1K (IN-1K) (Deng et al., 2009), object detection on MS COCO (Lin et al., 2015), and semantic segmentation on ADE20K (Zhou et al., 2017). Furthermore, to strictly evaluate the resilience of learned representations against distribution shifts and corruptions, we extend our evaluation to three challenging robustness benchmarks: ImageNet Corruption (IN-C) (Hendrycks & Dietterich, 2019), ImageNet Rendition (IN-R) (Hendrycks et al., 2021a), and ImageNet Adversarial (IN-A) (Hendrycks et al., 2021b). All experiments follow the same training schedules and evaluation protocols as in prior MambaVision work.

**Results.** Tables 4 and 5 summarize the results, where MuonSSM yields consistent improvements over the MambaVision baseline across standard classification and dense prediction tasks. Crucially, in the robustness evaluation, MuonSSM demonstrates a significant reduction in Mean Corruption Error on IN-C and higher accuracy on IN-A and

*Table 2.* Zero-shot performance on common sense reasoning and language modeling tasks. All models are pretrained on FineWeb-Edu10B tokens. We compare original SSM backbones with our **Muon**-integrated versions. ↓: Lower is better, ↑: Higher is better. **Bold** indicates the best result per architecture.

| Architecture | Memory Algorithm | Wiki. ppl ↓ | LMB. ppl ↓ | LMB. acc ↑ | PIQA acc ↑ | Hella. acc_n ↑ | Wino. acc ↑ | ARC-e acc ↑ | ARC-c acc_n ↑ | SIQA acc ↑ | BoolQ acc ↑ | Avg. acc ↑ |
|---|---|---|---|---|---|---|---|---|---|---|---|---|
| Mamba | Original | 42.17 | 102.51 | 20.57 | 63.81 | 30.10 | 51.11 | 52.39 | 21.75 | 37.41 | 59.87 | 42.13 |
| | + Muon (Ours) | **40.83** | **89.17** | **22.84** | 63.47 | **33.19** | **53.36** | **53.21** | **25.58** | **38.33** | **63.82** | **44.23** |
| LongHorn | Original | 43.06 | 96.80 | 22.16 | 62.79 | 29.87 | 52.25 | 50.34 | 21.24 | 36.18 | 55.02 | 41.23 |
| | + Muon (Ours) | **41.71** | **80.98** | **24.00** | 62.02 | **32.85** | **54.38** | **51.17** | **25.12** | **37.15** | **59.44** | **43.27** |
| Gated DeltaNet | Original | 39.58 | 97.92 | 21.36 | 62.45 | 30.02 | 51.30 | 51.85 | 21.42 | 36.90 | 55.23 | 41.32 |
| | + Muon (Ours) | **38.12** | **83.47** | **23.91** | **62.88** | **33.51** | **53.76** | **52.74** | **23.13** | **38.02** | **56.85** | **43.10** |

*Table 3.* Needle-in-a-Haystack retrieval performance under varying context lengths. We evaluate three variants: PassKey (PK), Number (N), and UUID. Higher values indicate better accuracy.

| Architecture | Memory Algorithm | S-NIAH-PK 2K | 4K | 8K | S-NIAH-N 2K | 4K | 8K | S-NIAH-UUID 2K | 4K | 8K |
|---|---|---|---|---|---|---|---|---|---|---|
| Mamba | Original | 29.3 | 16.4 | 8.8 | 18.6 | 14.3 | 4.1 | 48.6 | 32.9 | 25.0 |
| | + Muon | **32.1** | **20.5** | **15.8** | **22.4** | **19.1** | **10.2** | **53.8** | **38.2** | **31.5** |
| LongHorn | Original | **67.9** | 52.1 | 20.0 | 70.7 | 55.7 | 35.6 | 46.4 | 30.7 | 19.3 |
| | + Muon | 66.7 | **55.9** | **39.3** | **75.1** | **71.4** | **36.8** | **52.9** | **37.9** | **28.6** |
| GatedDeltaNet | Original | 61.4 | 43.6 | 25.7 | 69.3 | 43.6 | 27.1 | 52.1 | 35.0 | 24.3 |
| | + Muon | **63.2** | **48.9** | **44.5** | **74.1** | **57.8** | **29.4** | **58.3** | **42.6** | **33.1** |

IN-R. This enhanced resilience indicates that the spectral whitening effect of MuonSSM prevents the model from overfitting to superficial high-frequency statistics or texture biases common in standard training. Instead, the optimization dynamics encourage the learning of invariant features robust to distribution shifts, proving beneficial for out-of-distribution generalization.

*Table 4.* Comparison of SSMs with and without the proposed Muon framework. **Bold** indicates the best performance per architecture.

| Architecture | Memory Algorithm | IN-1K Top-1 ↑ | Top-5 ↑ | IN-R Top-1 ↑ | IN-A Top-1 ↑ | IN-C Top-1 ↑ | mCE ↓ |
|---|---|---|---|---|---|---|---|
| Mamba | Original | 81.08 | 95.32 | 42.35 | 20.57 | 12.31 | 112.84 |
| | + Muon | **81.19** | **95.36** | **42.61** | 20.50 | **12.57** | **112.52** |
| LongHorn | Original | 81.63 | 95.82 | 45.44 | 23.76 | 13.12 | 111.68 |
| | + Muon | **82.01** | **95.90** | **46.28** | **25.27** | **13.53** | **111.24** |
| GatedDeltaNet | Original | 79.92 | 95.24 | 41.55 | 19.92 | 11.85 | 114.12 |
| | + Muon | **80.31** | **95.35** | **42.18** | **20.47** | **12.33** | **113.56** |

*Table 5.* Downstream task performance on COCO2017 (Object Detection & Instance Segmentation) and ADE20K (Semantic Segmentation). We compare the original backbones against our **Muon**-integrated versions across different model scales.

| Architecture | Memory Algorithm | Object Detection $AP^{box}$ | $AP^{box}_{50}$ | $AP^{box}_{75}$ | Instance Seg. $AP^{mask}$ | $AP^{mask}_{50}$ | $AP^{mask}_{75}$ | Sem. Seg. mIoU |
|---|---|---|---|---|---|---|---|---|
| Mamba | Original | 50.8 | 69.5 | 55.2 | 44.1 | 67.0 | 47.9 | 43.9 |
| | + Muon | **51.1** | **69.9** | **55.4** | **44.3** | **67.4** | **48.2** | **45.2** |
| LongHorn | Original | 50.6 | 69.3 | 55.3 | 44.0 | 66.7 | 47.6 | 44.2 |
| | + Muon | **51.0** | **69.8** | **55.4** | **44.1** | **67.1** | 47.6 | **45.7** |
| GatedDeltaNet | Original | 49.5 | 68.1 | 53.8 | 43.4 | 67.2 | 46.8 | 41.2 |
| | + Muon | **50.1** | **68.8** | **54.5** | **43.8** | **67.8** | **47.3** | **41.8** |

### 4.3. Time-Series for Human Activity Recognition

*Table 6.* Comparison of different architectures with Memory Algorithm column added

| Architecture | Memory Algorithm | Accuracy (%) | Precision (%) | Recall (%) | F1-score (%) |
|---|---|---|---|---|---|
| **MuWiGes** | | | | | |
| Mamba | Original | 96.25 | 96.31 | 96.24 | 96.25 |
| | + Muon | **97.64** | **97.47** | **97.44** | **97.45** |
| LongHorn | Original | 97.23 | 97.44 | 97.27 | 97.35 |
| | + Muon | **97.95** | **97.98** | **97.95** | **97.96** |
| GatedDeltaNet | Original | 96.88 | 96.90 | 96.87 | 96.87 |
| | + Muon | **97.73** | **97.75** | **97.72** | **97.73** |
| **UESTC-MMEA-CL** | | | | | |
| Mamba | Original | 87.74 | 88.38 | 89.46 | 88.91 |
| | + Muon | **91.62** | **91.41** | **90.95** | **90.96** |
| LongHorn | Original | 89.06 | 89.88 | 89.44 | 89.43 |
| | + Muon | **91.56** | **90.94** | **91.12** | **91.02** |
| GatedDeltaNet | Original | 86.09 | 87.20 | 86.16 | 85.92 |
| | + Muon | **87.97** | **88.35** | **87.82** | **87.81** |
| **MMAct** | | | | | |
| Mamba | Original | 71.46 | 71.82 | 71.56 | 71.68 |
| | + Muon | **74.65** | **78.16** | **74.06** | **74.25** |
| LongHorn | Original | 72.47 | 75.68 | 74.12 | 73.76 |
| | + Muon | **74.40** | **79.25** | **76.47** | **76.43** |
| GatedDeltaNet | Original | 66.39 | 73.23 | 68.28 | 67.75 |
| | + Muon | **66.61** | **74.11** | **69.09** | **68.73** |

**Setup.** We adopt a simple architecture consisting of a Conv1D front-end for local feature extraction, followed by stacked state space blocks and a classification head. Within this architecture, standard SSM blocks are replaced with MuonSSM blocks, while all other components remain unchanged. This design ensures that any observed differences can be attributed to the memory update dynamics introduced by MuonSSM. We evaluate on three widely used benchmarks: MuWiGes (Nguyen et al., 2023) with 12 fine-grained hand gestures, UESTC-MMEA-CL (Xu et al., 2024) with 32 daily activities, and MMAct (Kong et al., 2019) with 37 complex actions with high sensor noise, covering diverse levels of activity granularity and temporal complexity.

**Results.** As shown in Table 6, MuonSSM outperforms baselines on all three datasets, with the margin of im-

provement widening as dataset complexity increases from MuWiGes to MMAct, which involve greater inter-class similarity and motion variability. Notably, these improvements arise despite the fact that the input sequences are not extremely long, suggesting that MuonSSM benefits temporal modeling by stabilizing update dynamics and reducing sensitivity to high-frequency noise, rather than solely by extending effective context length.

## 5. Empirical Analysis

### 5.1. Spectral Conditioning and Stability

Our core hypothesis is that standard first-order SSM updates suffer from spectral degradation over time, leading to ill-conditioned optimization landscapes. To verify this, we visualize the distribution of singular values of the state transition matrix ($d_{state} = 64$) for a Mamba backbone trained on the Human Activity Recognition task.

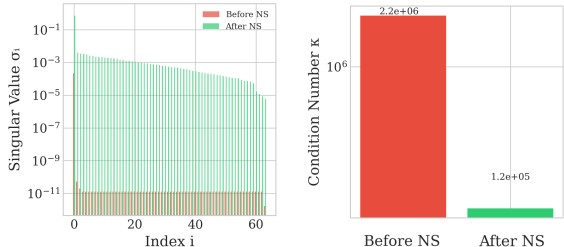

*Figure 2.* Singular value spectrum of the recurrent state matrix. Standard SSM updates (red) exhibit spectral collapse and a high condition number, while MuonSSM (green) maintains a near-isometric, well-conditioned spectrum via Newton–Schulz normalization.

As shown in Figure 2, the baseline Mamba model (red) exhibits a severe spectral collapse, where a few dominant singular values absorb most of the energy, while the majority decay towards zero. This results in an extremely high condition number ($\kappa \approx 2.2 \times 10^6$), indicating a brittle optimization landscape prone to vanishing gradients. In contrast, applying MuonSSM with Newton–Schulz iteration (green) effectively flattens the spectrum, pushing smaller singular values towards unity. This reduces the condition number by approximately $18\times$ (from $\kappa \approx 2.2 \times 10^6$ to $\kappa \approx 1.2 \times 10^5$). This empirical evidence confirms that our lightweight orthogonalization acts as an effective preconditioner, maintaining a healthy effective rank throughout training without the cost of exact SVD.

### 5.2. Optimization Dynamics and Computational Scalability

**Convergence Speed.** Figure 3a compares the pre-training loss on FineWeb-Edu 10B for the baseline LongHorn and our MuonSSM. The results are striking, LongHorn with Muon not only achieves a lower final perplexity but also demonstrates a significantly steeper initial learning trajec-

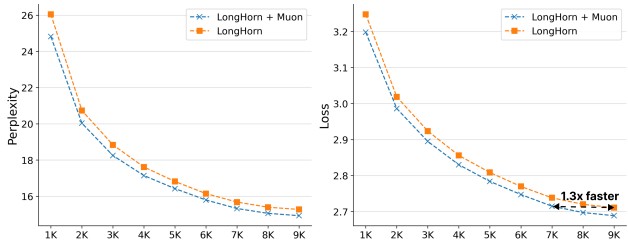

*(a)* Pre-training dynamics on FineWeb-Edu 10B tokens

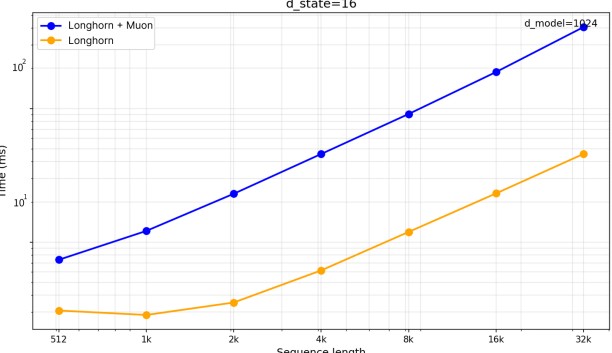

*(b)* Training time per epoch vs. sequence length.

*Figure 3.* (a) MuonSSM accelerates convergence $1.3\times$ faster, achieves lower validation loss and perplexity compared to the baseline. (b) The parallel trend lines indicate that MuonSSM preserves the asymptotic linear complexity of the baseline, adding only a constant-factor overhead due to the projection step.

tory. This suggests that the geometry-aware updates allow the optimizer to escape saddle points more efficiently and navigate the loss landscape with greater stability, effectively accelerating convergence by approximately $1.3\times$ in terms of training steps.

**Computational Scalability.** Integrating Newton–Schulz iterations introduces additional matrix multiplications, which may affect throughput. To assess this cost, we measure the training time per epoch across varying sequence lengths on the MMAct dataset. As illustrated in Figure 3b, LongHorn with Muon incurs a modest constant-factor overhead compared to the baseline LongHorn, while exhibiting a similar scaling trend as sequence length increases. This indicates that MuonSSM preserves the scan-compatible structure of the underlying SSM backbone: the recurrence can still be evaluated with $\mathcal{O}(\log L)$ parallel depth and $\mathcal{O}(L)$ total work over the sequence.

Although the per-step computation is slightly higher, the faster convergence rate means that MuonSSM can reach target performance targets in fewer total steps, making it a highly efficient strategy overall.

### 5.3. Capacity vs. Geometric Conditioning

The augmented state $\mathcal{Z}_t = [S_t \ M_t] \in \mathbb{R}^{d \times 2m}$ (Proposition 3.1) effectively doubles the memory dimension, raising the question of whether the observed gains can be attributed

solely to increased capacity.

To isolate this effect, we construct $2 \times d_{state}$ baselines for both LongHorn and Mamba by doubling the state dimension while keeping all other components unchanged. All models are trained on MMAct over 5 independent runs. Table 7 reports accuracy, precision, recall, and F1 scores for these variants.

Doubling $d_{\text{state}}$ yields only modest improvements over the base models ($+0.41\%$ for LongHorn, $+1.05\%$ for Mamba). In contrast, Muon variants outperform their respective $2\times$ baselines by a substantial margin ($+1.52\%$ and $+2.13\%$), indicating that increased capacity alone is insufficient to explain the gains. These results support our hypothesis that the primary benefit of MuonSSM arises from improved optimization geometry, rather than from a simple expansion of the state dimension.

### 5.4. Ablation Studies: Impact of Newton–Schulz Iterations.

A critical design choice in MuonSSM is the number of Newton–Schulz iterations. In Figure 4, we compare Mamba baselines against MuonSSM variants with different settings on the MMAct dataset: Momentum Only (No NS), Newton–Schulz one iteration, and Newton–Schulz five iterations. This suggests that enforcing strict via more iterations may be overly rigid, potentially hindering the model's ability to capture necessary non-orthogonal correlations in the data. Based on these results, we adopt Newton–Schulz with one iteration as the default setting, balancing performance with computational efficiency.

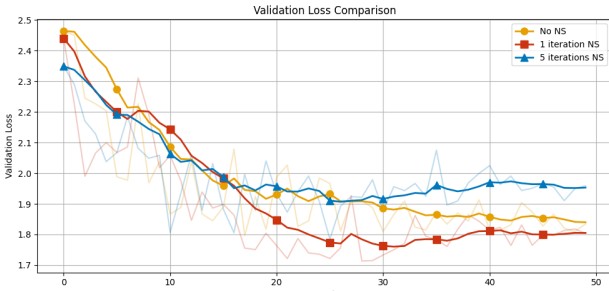

*Figure 4.* Ablation of Iterations on MMAct dataset

### 5.5. Robustness and Inductive Bias

Finally, we investigate *what* features MuonSSM learns compared to standard models. We employ GradCAM (Gildenblat & contributors, 2021) to visualize the focus regions of MambaVision-Tiny models on the IN-R dataset, which contains out-of-distribution examples like art, sketches, and cartoons. Figure 5 reveals a striking difference in inductive bias.

**Texture vs. Shape Bias.** The baseline MambaVision (left) often misclassifies objects based on texture cues. For in-

stance, it misidentifies a sketch of a *Hammerhead Shark* as a *Great White Shark*, likely due to texture ambiguity, and mistakes a stylized *Goldfish* for an *Old English Sheepdog*.

**Invariant Representation.** MuonSSM (right) correctly classifies both instances. The activation maps show that MuonSSM focuses more precisely on the structural shape of the distinct head of the hammerhead shark rather than background noise or texture. This suggests that the spectral orthogonalization in MuonSSM encourages the learning of more disentangled and shape-invariant features, leading to the superior robustness observed in our main experiments.

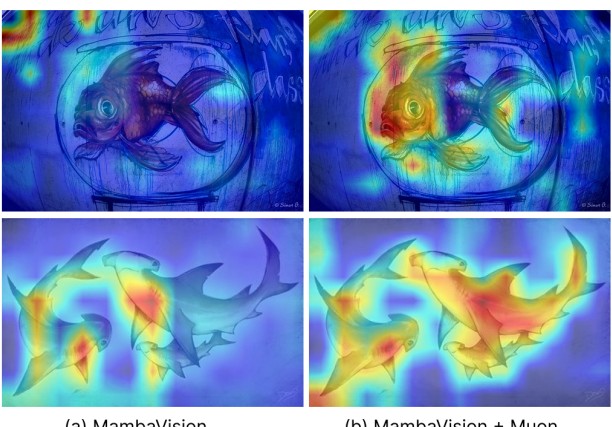

(a) MambaVision          (b) MambaVision + Muon

*Figure 5.* GradCAM visualizations show that standard MambaVision (left) is prone to texture bias and confusion under domain shift. MuonSSM (right) demonstrates stronger shape bias, correctly identifying the *Goldfish* and *Hammerhead Shark* by focusing on relevant structural features despite the artistic rendition.

## 6. Related Work

MuonSSM relates to prior work on state space sequence models, associative memory mechanisms, momentum-based recurrent dynamics, and orthogonalized updates. Unlike approaches that modify the recurrent transition operator, MuonSSM conditions the geometry of input-dependent memory updates while preserving scan-based parallel computation.

**State Space Models.** SSMs have emerged as scalable alternatives to attention for long-sequence modeling, beginning with structured models such as S4 (Gu et al., 2021) and extending to H3 (Fu et al., 2022), S5 (Smith et al., 2022), and Mamba (Gu & Dao, 2024; Dao & Gu, 2024). Their foundations are closely related to HiPPO-based memory compression (Gu et al., 2020; 2022b) and subsequent diagonal or parameterized SSM variants (Gu et al., 2022a). Related linear-time architectures combine recurrence with limited attention or alternative sequence mixing mechanisms, including Mega (Ma et al., 2022), Griffin (De et al., 2024), Jamba (Lenz et al., 2025), RWKV (Peng et al., 2023), RetNet (Sun et al., 2023), and Hyena (Poli et al., 2023). Across these models, efficiency is typically achieved by structuring

*Table 7.* Comparison with doubled state-dimension baselines on MMAct (5 independent runs, mean $\pm$ std). Best results are in **bold**.

| Architecture | Accuracy (%) | Precision (%) | Recall (%) | F1 (%) |
|---|---|---|---|---|
| LongHorn | $72.47 \pm 0.19$ | $75.68 \pm 0.30$ | $74.12 \pm 0.17$ | $73.76 \pm 0.24$ |
| LongHorn $2 \times d_{\text{state}}$ | $72.88 \pm 0.27$ | $78.62 \pm 0.34$ | $74.67 \pm 0.25$ | $74.90 \pm 0.23$ |
| **MuonLongHorn (Ours)** | $\mathbf{74.40 \pm 0.21}$ | $\mathbf{79.25 \pm 0.26}$ | $\mathbf{76.47 \pm 0.27}$ | $\mathbf{76.43 \pm 0.36}$ |
| Mamba | $71.47 \pm 0.25$ | $71.82 \pm 0.38$ | $71.56 \pm 0.28$ | $71.68 \pm 0.33$ |
| Mamba $2 \times d_{\text{state}}$ | $72.52 \pm 0.22$ | $76.98 \pm 0.34$ | $73.80 \pm 0.25$ | $73.40 \pm 0.18$ |
| **MuonMamba (Ours)** | $\mathbf{74.65 \pm 0.34}$ | $\mathbf{78.16 \pm 0.29}$ | $\mathbf{74.06 \pm 0.41}$ | $\mathbf{74.25 \pm 0.35}$ |

the recurrence for parallel scan primitives (Blelloch, 1990). Recent work further connects SSMs, linear attention, and recurrent computation through low-rank associative memory updates (Katharopoulos et al., 2020; Orvieto et al., 2023; Behrouz et al., 2025b). Several models interpret state updates as associative memory rules or online learning dynamics (Yang et al., 2024b;a; Liu et al., 2024; Behrouz et al., 2025a; Ba et al., 2016; Ramsauer et al., 2020; Schlag et al., 2021), but largely retain first-order update mechanisms. Beyond this regime, LinOSS (Rusch & Rus, 2024) derives stable second-order dynamics from forced harmonic oscillators, yielding a linear time-invariant formulation. In contrast, MuonSSM targets selective, input-dependent SSMs, where transitions vary across timesteps and stability is addressed through per-step spectral conditioning of low-rank memory injections.

**Momentum in Sequence Models.** Momentum is a classical mechanism for stabilizing optimization and accelerating convergence (Polyak, 1964). It has also been explored in recurrent architectures as a way to extend effective memory and improve stability (Nguyen et al., 2020; Ma et al., 2022; Teo & Nguyen, 2024). Related perspectives connect recurrent computation with online optimization and test-time memory updates (Zinkevich, 2003; Behrouz et al., 2025b; 2024). MuonSSM builds on these ideas by incorporating momentum directly into scan-compatible SSM updates, yielding an additional pathway for information and gradient propagation without breaking the affine recurrence structure.

**Orthogonalization and Spectral Conditioning.** Another line of work stabilizes recurrent or deep networks through spectral constraints, unitary or orthogonal parameterizations, and normalization-based control of singular values (Pascanu et al., 2013; Arjovsky et al., 2016; Henaff et al., 2016; Vorontsov et al., 2017; Wisdom et al., 2016; Helfrich et al., 2018; Miyato et al., 2018). Exact orthogonalization is often costly, motivating lightweight alternatives based on Newton–Schulz iterations and related approximate normalization schemes (Song et al., 2022; Bernstein & Newhouse, 2024; Jordan et al., 2024).Recent optimizers such as Muon (Jordan et al., 2024) and Dion (Ahn et al., 2025) demonstrate the practical value of orthonormalized update directions for

large-scale training. MuonSSM differs from these optimizer-level methods: it applies lightweight Newton–Schulz-style conditioning directly to input-dependent low-rank memory injections inside scan-based SSMs, rather than orthogonalizing parameter gradients or imposing global constraints on the recurrent transition operator.

## 7. Concluding Remarks

In this work, we presented MuonSSM, a family of SSMs that stabilizes sequence modeling by conditioning memory updates rather than constraining recurrent transitions. MuonSSM combines a momentum pathway with a lightweight single-step Newton–Schulz transformation on low-rank input injections, yielding bounded and better-conditioned updates while preserving parallel scan complexity. Our analysis shows that this design provides an additional gradient pathway, controls spectral amplification, and encourages richer memory representations. Experiments across language, vision, and time-series tasks demonstrate consistent gains across multiple SSM backbones. While our study focuses on moderate-scale models and fixed conditioning hyperparameters, the results suggest that geometric conditioning of update dynamics is a simple and effective mechanism for stable sequence modeling. Future work will explore larger-scale pretraining, hybrid attention–SSM architectures, and adaptive conditioning strategies.

## Impact Statement

This paper presents work whose goal is to advance the field of Machine Learning. There are many potential societal consequences of our work, none which we feel must be specifically highlighted here.

## Acknowledgements

This research is supported by the National Research Foundation Singapore under the AI Singapore Programme (AISG Award No: AISG2-TC-2023-012-SGIL). This research is also supported by the Ministry of Education, Singapore, under the Academic Research Fund Tier 1 (FY2023) (A-8002040-00-00, A-8002039-00-00), the NUS Presidential Young Professorship Award (A-0009807-01-00), the NUS Artificial Intelligence Institute–Seed Funding (A-8003062-00-00), and the Cross Faculty Grant 2025, CFG25-012 (A-8004460-00-00).

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

# Supplement to "MuonSSM: Orthogonalizing State Space Models for Sequence Modeling"

## Appendix A: Proofs of Theoretical Results

In this appendix, we provide detailed proofs for all theoretical results stated in Section 3, along with additional lemmas and technical details.

### A.1. Proof of Proposition 3.1 and Remark 3.2: Parallelizability

*Proof.* To establish the parallelizability of MuonSSM, we provide a two-part proof. First, we verify that the proposed block-affine recurrence exactly recovers the coupled memory and momentum dynamics. Second, we prove the associativity of the transition operator, which is the sufficient condition for applying parallel associative scans.

**1. Recovery of Coupled Dynamics.** Given the augmented state $\mathcal{Z}_t = \begin{bmatrix} \mathbf{S}_t & \mathbf{M}_t \end{bmatrix}$, the block-affine recurrence $\mathcal{Z}_t = \mathcal{Z}_{t-1}\mathbf{\Phi}_t + \mathbf{\Psi}_t$ can be expanded using the definitions of $\mathbf{\Phi}_t$ and $\mathbf{\Psi}_t$:

$$\begin{bmatrix} \mathbf{S}_t & \mathbf{M}_t \end{bmatrix} = \begin{bmatrix} \mathbf{S}_{t-1} & \mathbf{M}_{t-1} \end{bmatrix} \begin{bmatrix} \mathbf{D}_t & \mathbf{0} \\ \gamma\mathbf{I} & \gamma\mathbf{I} \end{bmatrix} + \begin{bmatrix} \mathbf{U}_t & \mathbf{U}_t \end{bmatrix}. \tag{14}$$

Performing the block matrix multiplication yields the following two coupled equations:

$$\mathbf{S}_t = \mathbf{S}_{t-1}\mathbf{D}_t + \gamma\mathbf{M}_{t-1} + \mathbf{U}_t, \tag{15}$$
$$\mathbf{M}_t = \gamma\mathbf{M}_{t-1} + \mathbf{U}_t. \tag{16}$$

Substituting Eq. (16) into Eq. (15), we obtain:

$$\mathbf{S}_t = \mathbf{S}_{t-1}\mathbf{D}_t + \mathbf{M}_t. \tag{17}$$

This confirms that the block-affine formulation exactly matches the intended MuonSSM momentum-augmented memory update.

**2. Associativity and Parallel Complexity.** For a sequence of affine transformations $f_t(\mathcal{Z}) = \mathcal{Z}\mathbf{\Phi}_t + \mathbf{\Psi}_t$, the composition of two successive steps $f_j(f_i(\mathcal{Z}))$ is:

$$\begin{aligned} f_j(f_i(\mathcal{Z})) &= (\mathcal{Z}\mathbf{\Phi}_i + \mathbf{\Psi}_i)\mathbf{\Phi}_j + \mathbf{\Psi}_j \\ &= \mathcal{Z}(\mathbf{\Phi}_i\mathbf{\Phi}_j) + (\mathbf{\Psi}_i\mathbf{\Phi}_j + \mathbf{\Psi}_j). \end{aligned}$$

This composition defines the operator $\oplus$:

$$(\mathbf{\Phi}_i, \mathbf{\Psi}_i) \oplus (\mathbf{\Phi}_j, \mathbf{\Psi}_j) = (\mathbf{\Phi}_i\mathbf{\Phi}_j, \mathbf{\Psi}_i\mathbf{\Phi}_j + \mathbf{\Psi}_j).$$

To show that $\oplus$ is associative, we consider three elements $a, b, c$ where $a = (\mathbf{\Phi}_1, \mathbf{\Psi}_1)$, $b = (\mathbf{\Phi}_2, \mathbf{\Psi}_2)$, and $c = (\mathbf{\Phi}_3, \mathbf{\Psi}_3)$. Evaluating $(a \oplus b) \oplus c$:

$$\begin{aligned} (a \oplus b) \oplus c &= (\mathbf{\Phi}_1\mathbf{\Phi}_2, \mathbf{\Psi}_1\mathbf{\Phi}_2 + \mathbf{\Psi}_2) \oplus (\mathbf{\Phi}_3, \mathbf{\Psi}_3) \\ &= ((\mathbf{\Phi}_1\mathbf{\Phi}_2)\mathbf{\Phi}_3, (\mathbf{\Psi}_1\mathbf{\Phi}_2 + \mathbf{\Psi}_2)\mathbf{\Phi}_3 + \mathbf{\Psi}_3) \\ &= (\mathbf{\Phi}_1\mathbf{\Phi}_2\mathbf{\Phi}_3, \mathbf{\Psi}_1\mathbf{\Phi}_2\mathbf{\Phi}_3 + \mathbf{\Psi}_2\mathbf{\Phi}_3 + \mathbf{\Psi}_3). \end{aligned}$$

Evaluating $a \oplus (b \oplus c)$:

$$\begin{aligned} a \oplus (b \oplus c) &= (\mathbf{\Phi}_1, \mathbf{\Psi}_1) \oplus (\mathbf{\Phi}_2\mathbf{\Phi}_3, \mathbf{\Psi}_2\mathbf{\Phi}_3 + \mathbf{\Psi}_3) \\ &= (\mathbf{\Phi}_1(\mathbf{\Phi}_2\mathbf{\Phi}_3), \mathbf{\Psi}_1(\mathbf{\Phi}_2\mathbf{\Phi}_3) + (\mathbf{\Psi}_2\mathbf{\Phi}_3 + \mathbf{\Psi}_3)) \\ &= (\mathbf{\Phi}_1\mathbf{\Phi}_2\mathbf{\Phi}_3, \mathbf{\Psi}_1\mathbf{\Phi}_2\mathbf{\Phi}_3 + \mathbf{\Psi}_2\mathbf{\Phi}_3 + \mathbf{\Psi}_3). \end{aligned}$$

Since $(a \oplus b) \oplus c = a \oplus (b \oplus c)$, the operator is associative. By the property of associative scans, any sequence of $L$ elements can be reduced in $\mathcal{O}(\log L)$ time using a binary tree structure. Given that the Newton–Schulz step for $\mathbf{U}_t$ is local to each timestep, the entire sequence $\mathcal{Z}_{1:L}$ is computable in $\mathcal{O}(\log L)$ parallel depth. $\square$

### A.2. Proof of Proposition 3.3: Gradient Stability

*Proof.* The result is derived by applying the chain rule to the backpropagation through time dynamics. Given the state evolution in row-vector convention $\mathcal{Z}_t = \mathcal{Z}_{t-1}\boldsymbol{\Phi}_t + \boldsymbol{\Psi}_t$, we evaluate the gradient of the total loss $\mathcal{L}$ with respect to the state at time $t-1$. Following the standard convention, the gradient relates to the final state at time $T$ via the product of transposed transition matrices:

$$\frac{\partial \mathcal{L}}{\partial \mathcal{Z}_{t-1}} = \frac{\partial \mathcal{L}}{\partial \mathcal{Z}_T} \left( \prod_{n=T}^{t} \frac{\partial \mathcal{Z}_n}{\partial \mathcal{Z}_{n-1}} \right) = \frac{\partial \mathcal{L}}{\partial \mathcal{Z}_T} \underbrace{\left( \prod_{n=t}^{T} \boldsymbol{\Phi}_n \right)^{\top}}_{\mathcal{J}_{(t-1)\to T}^{\top}}, \tag{18}$$

where $\frac{\partial \mathcal{Z}_n}{\partial \mathcal{Z}_{n-1}} = \boldsymbol{\Phi}_n^{\top}$. Recalling the block structure from A.1, the transposed matrix is:

$$\boldsymbol{\Phi}_n^{\top} = \begin{bmatrix} \mathbf{D}_n & \mathbf{0} \\ \gamma\mathbf{I} & \gamma\mathbf{I} \end{bmatrix}^{\top} = \begin{bmatrix} \mathbf{D}_n^{\top} & \gamma\mathbf{I} \\ \mathbf{0} & \gamma\mathbf{I} \end{bmatrix}.$$

The product $\mathcal{J}_{(t-1)\to T}^{\top}$ involves multiplying these upper-triangular matrices. Let $\mathcal{J}_{(t-1)\to T}^{\top} = \begin{bmatrix} \mathbf{A} & \mathbf{B} \\ \mathbf{0} & \mathbf{C} \end{bmatrix}$, where the blocks are determined by induction over the sequence length $T - t + 1$:

**1. Diagonal Blocks:** The top-left block $\mathbf{A}$ accumulates the sequence of memory transitions: $\mathbf{A} = \prod_{n=T}^{t} \mathbf{D}_n^{\top}$. The bottom-right block $\mathbf{C}$ represents the persistent momentum decay: $\mathbf{C} = \prod_{n=T}^{t} \gamma\mathbf{I} = (\gamma\mathbf{I})^{T-t+1}$.

**2. Off-diagonal Momentum Block:** The block $\mathbf{B}$ captures the cross-propagation from the momentum pathway into the memory state. Expanding the product, the top-right block accumulates terms where each momentum injection $\gamma\mathbf{I}$ at step $k$ is subsequently transformed by the memory transitions $\mathbf{D}_j^{\top}$. This yields:

$$\mathbf{B} = \sum_{k=t}^{T} \left( \prod_{j=T}^{k+1} \mathbf{D}_j^{\top} \right) (\gamma\mathbf{I})^{k-t+1}.$$

As $\gamma \to 1$, the bottom-right block $\mathbf{C}$ approaches the identity matrix, ensuring that a non-vanishing component of the gradient $\frac{\partial \mathcal{L}}{\partial \mathbf{M}_T}$ is preserved and propagated back to $\mathbf{M}_{t-1}$, regardless of the contractive nature of the memory transitions $\mathbf{D}_n$. Substituting these blocks back into $\mathcal{J}_{(t-1)\to T}^{\top}$ completes the proof. $\square$

### A.3. Proof of Corollary 3.5: Spectral Conditioning

*Proof.* **1. Spectral Transformation.** Let $\tilde{X} = X / \max(\|X\|_F, \delta)$, so $\|\tilde{X}\|_F \leq 1$. By the standard inequality $\|\cdot\|_2 \leq \|\cdot\|_F$, all singular values of $\tilde{X}$ satisfy $\sigma_i \in [0, 1]$. Let $\tilde{X} = U\Sigma V^{\top}$ be the SVD. Since $\tilde{X}\tilde{X}^{\top} = U\Sigma^2 U^{\top}$, the NS update satisfies:

$$\text{NS}(X) = \left( a + b\,\tilde{X}\tilde{X}^{\top} + c\,(\tilde{X}\tilde{X}^{\top})^2 \right)\tilde{X} = U \underbrace{\left( a\Sigma + b\Sigma^3 + c\Sigma^5 \right)}_{= \rho(\Sigma)} V^{\top},$$

where $\rho(\sigma) = a\sigma + b\sigma^3 + c\sigma^5$ is applied entry-wise. Thus $\sigma_{\max}(\text{NS}(X)) = \max_i \rho(\sigma_i) \leq \sup_{\sigma \in [0,1]} \rho(\sigma)$, and it suffices to analyze $\rho$ on $[0, 1]$.

**2. Global Extremum on $[0, 1]$.** The derivative is:

$$\rho'(\sigma) = 3.4445 - 14.325\,\sigma^2 + 10.1575\,\sigma^4.$$

Setting $\rho'(\sigma) = 0$ and substituting $u = \sigma^2$ yields:

$$10.1575\,u^2 - 14.325\,u + 3.4445 = 0, \qquad \Delta = 14.325^2 - 4(10.1575)(3.4445) \approx 65.26 > 0.$$

The two roots are:

$$u_{1,2} = \frac{14.325 \pm \sqrt{65.26}}{2 \times 10.1575} \approx \{0.3075,\ 1.1028\}.$$

Only $u_1 \approx 0.3075$ lies in $[0, 1]$, giving the unique interior critical point $\sigma^* = \sqrt{u_1} \approx 0.5545$. Since $\rho''(\sigma^*) = -28.65\,\sigma^* + 40.63\,(\sigma^*)^3 \approx -8.9593 < 0$, this is a *local maximum*. Evaluating $\rho$ at all candidates:

$$\rho(0) = 0,$$
$$\rho(\sigma^*) = 3.4445(0.5545) - 4.7750(0.5545)^3 + 2.0315(0.5545)^5 \approx 1.2,$$
$$\rho(1) = 3.4445 - 4.7750 + 2.0315 = 0.701$$

Therefore:

$$\sup_{\sigma \in [0,1]} \rho(\sigma) = \rho(\sigma^*) \approx 1.2,$$

which gives $\sigma_{\max}(\mathrm{NS}(X)) \lesssim 1.2 = 1 + \varepsilon_u$

**3. Preservation of rank-1 structure.** Factoring $\rho(\sigma) = \sigma \cdot q(\sigma^2)$ where $q(u) = cu^2 + bu + a$. The discriminant of $q$:

$$\Delta_q = b^2 - 4ac = (4.7750)^2 - 4(3.4445)(2.0315) = 22.801 - 27.985 = -5.189 < 0.$$

Since $\Delta_q < 0$ and $c > 0$, we have $q(u) > 0$ for all $u \in \mathbb{R}$. Therefore:

$$\rho(\sigma) > 0 \ \ \forall\, \sigma > 0, \qquad \rho(0) = 0.$$

It follows that $\mathrm{rank}(\mathrm{NS}(X)) = \mathrm{rank}(\widetilde{X}) = \mathrm{rank}(X)$, preserving rank-1 structure.

$\square$

### A.4. Proof of Proposition 3.6: Backward Geometry of Newton–Schulz Normalization

*Proof.* We prove the result for the normalized Newton–Schulz map in Definition 2.1, viewed as a function of the raw input $\mathbf{X}$. Since $\|\mathbf{X}_0\|_F = 1$, and in practice $\delta < 1$, the normalization is locally given by $\widetilde{\mathbf{X}} = \mathbf{X}/\|\mathbf{X}\|_F$ around $\mathbf{X}_0$.

Write

$$\mathcal{G}(\mathbf{X}) = P(\widetilde{\mathbf{X}})\widetilde{\mathbf{X}}, \qquad P(\mathbf{Z}) = a\mathbf{I}_d + b\mathbf{Z}\mathbf{Z}^\top + c(\mathbf{Z}\mathbf{Z}^\top)^2.$$

Let $\mathbf{X}_0 = \mathbf{u}\mathbf{w}^\top$ with $\|\mathbf{u}\|_2 = \|\mathbf{w}\|_2 = 1$.

**1. Differential of the Frobenius normalization.** For a perturbation $\mathbf{H} \in \mathbb{R}^{d \times m}$, the differential of $\widetilde{\mathbf{X}} = \mathbf{X}/\|\mathbf{X}\|_F$ at $\mathbf{X}_0$ is

$$d\widetilde{\mathbf{X}}[\mathbf{H}] = \mathbf{H} - \langle \mathbf{X}_0, \mathbf{H}\rangle_F \mathbf{X}_0 = \mathbf{H} - (\mathbf{u}^\top \mathbf{H}\mathbf{w})\mathbf{u}\mathbf{w}^\top. \tag{19}$$

Define

$$\alpha_{\mathbf{H}} := \mathbf{u}^\top \mathbf{H}\mathbf{w}, \qquad \mathbf{K} := d\widetilde{\mathbf{X}}[\mathbf{H}] = \mathbf{H} - \alpha_{\mathbf{H}}\mathbf{u}\mathbf{w}^\top.$$

Then

$$\mathbf{u}^\top \mathbf{K}\mathbf{w} = \alpha_{\mathbf{H}} - \alpha_{\mathbf{H}} = 0.$$

**2. Differential of the polynomial part.** At $\mathbf{X}_0 = \mathbf{u}\mathbf{w}^\top$, we have

$$\widetilde{\mathbf{X}}_0\widetilde{\mathbf{X}}_0^\top = \mathbf{u}\mathbf{u}^\top, \qquad (\widetilde{\mathbf{X}}_0\widetilde{\mathbf{X}}_0^\top)^2 = \mathbf{u}\mathbf{u}^\top.$$

Thus

$$P(\widetilde{\mathbf{X}}_0) = a\mathbf{I}_d + (b + c)\mathbf{u}\mathbf{u}^\top.$$

Let

$$\mathbf{A} = \widetilde{\mathbf{X}}\widetilde{\mathbf{X}}^\top.$$

The differential of $\mathbf{A}$ at $\widetilde{\mathbf{X}}_0$ in the direction $\mathbf{K}$ is

$$d\mathbf{A}[\mathbf{K}] = \mathbf{K}\mathbf{w}\mathbf{u}^\top + \mathbf{u}(\mathbf{K}\mathbf{w})^\top.$$

Using $\mathbf{u}^\top \mathbf{K}\mathbf{w} = 0$, the differential of $\mathbf{A}^2$ is

$$d(\mathbf{A}^2)[\mathbf{K}] = d\mathbf{A}[\mathbf{K}]\mathbf{u}\mathbf{u}^\top + \mathbf{u}\mathbf{u}^\top d\mathbf{A}[\mathbf{K}]$$
$$= \mathbf{K}\mathbf{w}\mathbf{u}^\top + \mathbf{u}(\mathbf{K}\mathbf{w})^\top.$$

Therefore,

$$dP[\mathbf{K}] = (b + c)\left(\mathbf{K}\mathbf{w}\mathbf{u}^\top + \mathbf{u}(\mathbf{K}\mathbf{w})^\top\right).$$

**3. Full Jacobian.** By the product rule,

$$DG_{\mathbf{X}_0}[\mathbf{H}] = dP[\mathbf{K}]\,\widetilde{\mathbf{X}}_0 + P(\widetilde{\mathbf{X}}_0)\mathbf{K}$$
$$= (b+c)\left(\mathbf{K}\mathbf{w}\mathbf{u}^\top + \mathbf{u}(\mathbf{K}\mathbf{w})^\top\right)\mathbf{u}\mathbf{w}^\top + a\mathbf{K} + (b+c)\mathbf{u}\mathbf{u}^\top\mathbf{K}. \tag{20}$$

Since

$$(\mathbf{K}\mathbf{w})^\top\mathbf{u} = \mathbf{u}^\top\mathbf{K}\mathbf{w} = 0,$$

the term $\mathbf{u}(\mathbf{K}\mathbf{w})^\top\mathbf{u}\mathbf{w}^\top$ vanishes. Hence

$$DG_{\mathbf{X}_0}[\mathbf{H}] = (b+c)\mathbf{K}\mathbf{w}\mathbf{w}^\top + a\mathbf{K} + (b+c)\mathbf{u}\mathbf{u}^\top\mathbf{K}. \tag{21}$$

Substituting $\mathbf{K} = \mathbf{H} - \alpha_{\mathbf{H}}\mathbf{u}\mathbf{w}^\top$ into Eq. (21) gives

$$DG_{\mathbf{X}_0}[\mathbf{H}] = a\mathbf{H} + (b+c)\mathbf{H}\mathbf{w}\mathbf{w}^\top + (b+c)\mathbf{u}\mathbf{u}^\top\mathbf{H} - \left(a + 2(b+c)\right)\alpha_{\mathbf{H}}\mathbf{u}\mathbf{w}^\top. \tag{22}$$

**4. Eigenvalue families.** We evaluate Eq. (22) on the four orthogonal subspaces induced by $\mathbf{u}$ and $\mathbf{w}$.

*Radial direction.* Let $\mathbf{H} = \mathbf{u}\mathbf{w}^\top$. Then $\alpha_{\mathbf{H}} = 1$, $\mathbf{H}\mathbf{w}\mathbf{w}^\top = \mathbf{H}$, and $\mathbf{u}\mathbf{u}^\top\mathbf{H} = \mathbf{H}$. Therefore,

$$DG_{\mathbf{X}_0}[\mathbf{H}] = \left(a + (b+c) + (b+c) - a - 2(b+c)\right)\mathbf{H} = 0.$$

Thus $\mathbf{u}\mathbf{w}^\top$ has eigenvalue 0.

*Directions $\mathbf{u}_\perp\mathbf{w}^\top$.* Let $\mathbf{H} = \mathbf{p}\mathbf{w}^\top$ with $\mathbf{p} \perp \mathbf{u}$. Then $\alpha_{\mathbf{H}} = 0$, $\mathbf{H}\mathbf{w}\mathbf{w}^\top = \mathbf{H}$, and $\mathbf{u}\mathbf{u}^\top\mathbf{H} = 0$. Hence

$$DG_{\mathbf{X}_0}[\mathbf{H}] = (a+b+c)\mathbf{H}.$$

Thus these directions have eigenvalue $a + b + c$.

*Directions $\mathbf{u}\mathbf{w}_\perp^\top$.* Let $\mathbf{H} = \mathbf{u}\mathbf{q}^\top$ with $\mathbf{q} \perp \mathbf{w}$. Then $\alpha_{\mathbf{H}} = 0$, $\mathbf{H}\mathbf{w}\mathbf{w}^\top = 0$, and $\mathbf{u}\mathbf{u}^\top\mathbf{H} = \mathbf{H}$. Hence

$$DG_{\mathbf{X}_0}[\mathbf{H}] = (a+b+c)\mathbf{H}.$$

Thus these directions also have eigenvalue $a + b + c$.

*Directions $\mathbf{u}_\perp\mathbf{w}_\perp^\top$.* Let $\mathbf{H} = \mathbf{p}\mathbf{q}^\top$ with $\mathbf{p} \perp \mathbf{u}$ and $\mathbf{q} \perp \mathbf{w}$. Then $\alpha_{\mathbf{H}} = 0$, $\mathbf{H}\mathbf{w}\mathbf{w}^\top = 0$, and $\mathbf{u}\mathbf{u}^\top\mathbf{H} = 0$. Hence

$$DG_{\mathbf{X}_0}[\mathbf{H}] = a\mathbf{H}.$$

Thus these directions have eigenvalue $a$.

The dimensions of the three eigenvalue families are respectively 1, $(d-1) + (m-1) = d + m - 2$, and $(d-1)(m-1)$, matching the statement of the proposition.

Finally, for Frobenius normalization, Eq. (19) shows that the radial direction $\mathbf{u}\mathbf{w}^\top$ is mapped to zero, while every direction orthogonal to $\mathbf{u}\mathbf{w}^\top$ is unchanged. Therefore its eigenvalues are 0 in the radial direction and 1 on all tangent directions. $\square$

### A.5. Proof of Proposition 3.8: Rank Enrichment

*Proof.* We prove each part separately.

**1. Momentum Expansion.** Expanding the recurrence

$$\mathbf{M}_t = \gamma\mathbf{M}_{t-1} + \mathrm{NS}(\tau\beta_t\mathbf{v}_t\mathbf{k}_t^\top)$$

with initial condition $\mathbf{M}_0 = \mathbf{0}$ gives

$$\mathbf{M}_1 = \mathrm{NS}(\tau\beta_1\mathbf{v}_1\mathbf{k}_1^\top),$$
$$\mathbf{M}_2 = \gamma\mathbf{M}_1 + \mathrm{NS}(\tau\beta_2\mathbf{v}_2\mathbf{k}_2^\top)$$
$$= \gamma\mathrm{NS}(\tau\beta_1\mathbf{v}_1\mathbf{k}_1^\top) + \mathrm{NS}(\tau\beta_2\mathbf{v}_2\mathbf{k}_2^\top),$$
$$\mathbf{M}_3 = \gamma\mathbf{M}_2 + \mathrm{NS}(\tau\beta_3\mathbf{v}_3\mathbf{k}_3^\top)$$
$$= \gamma^2\mathrm{NS}(\tau\beta_1\mathbf{v}_1\mathbf{k}_1^\top) + \gamma\mathrm{NS}(\tau\beta_2\mathbf{v}_2\mathbf{k}_2^\top) + \mathrm{NS}(\tau\beta_3\mathbf{v}_3\mathbf{k}_3^\top).$$

Continuing this expansion, or equivalently by induction, we obtain

$$\mathbf{M}_t = \sum_{s=1}^{t} \gamma^{t-s}\mathrm{NS}(\tau\beta_s\mathbf{v}_s\mathbf{k}_s^\top).$$

This proves Eq. (11).

**2. Rank Upper Bound.** Each input injection

$$\mathbf{X}_s = \tau \beta_s \mathbf{v}_s \mathbf{k}_s^\top$$

has rank at most one. Moreover, the Newton–Schulz map in Definition 2.1 acts on the singular values of $\mathbf{X}_s$ while preserving its singular directions. Hence

$$\mathrm{rank}\big(\mathrm{NS}(\mathbf{X}_s)\big) \leq \mathrm{rank}(\mathbf{X}_s) \leq 1.$$

Using the sub-additivity of matrix rank,

$$\mathrm{rank}(\mathbf{M}_t) \leq \sum_{s=1}^{t} \mathrm{rank}\Big(\mathrm{NS}(\tau \beta_s \mathbf{v}_s \mathbf{k}_s^\top)\Big) \leq t.$$

Since $\mathbf{M}_t \in \mathbb{R}^{d \times m}$, its rank is also bounded by $\min(d, m)$. Therefore,

$$\mathrm{rank}(\mathbf{M}_t) \leq \min(t, d, m).$$

This proves Eq. (12).

**3. Direction-Preserving Form of NS on Rank-1 Inputs.** We next make explicit how the Newton–Schulz operator acts on each rank-1 update. Under the non-degeneracy assumptions

$$\tau > 0, \qquad \beta_s > 0, \qquad \mathbf{v}_s \neq \mathbf{0}, \qquad \mathbf{k}_s \neq \mathbf{0},$$

define

$$\hat{\mathbf{v}}_s = \frac{\mathbf{v}_s}{\|\mathbf{v}_s\|_2}, \qquad \hat{\mathbf{k}}_s = \frac{\mathbf{k}_s}{\|\mathbf{k}_s\|_2},$$

and

$$\hat{\sigma}_s = \frac{\tau \beta_s \|\mathbf{v}_s\|_2 \|\mathbf{k}_s\|_2}{\max(\tau \beta_s \|\mathbf{v}_s\|_2 \|\mathbf{k}_s\|_2, \delta)} \in (0, 1].$$

Then the normalized input satisfies

$$\widetilde{\mathbf{X}}_s = \hat{\sigma}_s \hat{\mathbf{v}}_s \hat{\mathbf{k}}_s^\top.$$

Since this matrix is rank-1, we have

$$\widetilde{\mathbf{X}}_s \widetilde{\mathbf{X}}_s^\top = \hat{\sigma}_s^2 \hat{\mathbf{v}}_s \hat{\mathbf{v}}_s^\top, \qquad \left(\widetilde{\mathbf{X}}_s \widetilde{\mathbf{X}}_s^\top\right)^2 = \hat{\sigma}_s^4 \hat{\mathbf{v}}_s \hat{\mathbf{v}}_s^\top.$$

Substituting these identities into Definition 2.1 gives

$$\begin{aligned}
\mathrm{NS}(\mathbf{X}_s) &= \big(a\mathbf{I}_d + b\hat{\sigma}_s^2 \hat{\mathbf{v}}_s \hat{\mathbf{v}}_s^\top + c\hat{\sigma}_s^4 \hat{\mathbf{v}}_s \hat{\mathbf{v}}_s^\top\big) \hat{\sigma}_s \hat{\mathbf{v}}_s \hat{\mathbf{k}}_s^\top \\
&= \big(a\hat{\sigma}_s + b\hat{\sigma}_s^3 + c\hat{\sigma}_s^5\big) \hat{\mathbf{v}}_s \hat{\mathbf{k}}_s^\top \\
&= \rho(\hat{\sigma}_s) \hat{\mathbf{v}}_s \hat{\mathbf{k}}_s^\top,
\end{aligned}$$

where

$$\rho(\sigma) = a\sigma + b\sigma^3 + c\sigma^5.$$

For the coefficients used in Definition 2.1, we have $\rho(\sigma) > 0$ for all $\sigma > 0$. Therefore, each non-degenerate NS-normalized update is a positive scalar multiple of the same rank-1 direction $\hat{\mathbf{v}}_s \hat{\mathbf{k}}_s^\top$.

Consequently, the momentum state can be written as

$$\mathbf{M}_t = \sum_{s=1}^{t} \lambda_s \hat{\mathbf{v}}_s \hat{\mathbf{k}}_s^\top, \qquad \lambda_s := \gamma^{t-s} \rho(\hat{\sigma}_s) > 0. \tag{23}$$

**4. Attainability of the Upper Bound.**   Let

$$r = \min(t, d, m).$$

We construct one explicit configuration for which $\mathrm{rank}(\mathbf{M}_t) = r$. Choose $\hat{\mathbf{v}}_1, \ldots, \hat{\mathbf{v}}_r$ to be orthonormal vectors in $\mathbb{R}^d$ and $\hat{\mathbf{k}}_1, \ldots, \hat{\mathbf{k}}_r$ to be orthonormal vectors in $\mathbb{R}^m$. If $t > r$, choose the remaining directions as

$$\hat{\mathbf{v}}_s = \hat{\mathbf{v}}_1, \qquad \hat{\mathbf{k}}_s = \hat{\mathbf{k}}_1, \qquad s = r+1, \ldots, t.$$

Then Eq. (23) becomes

$$\mathbf{M}_t = \mu_1 \hat{\mathbf{v}}_1 \hat{\mathbf{k}}_1^\top + \sum_{s=2}^{r} \lambda_s \hat{\mathbf{v}}_s \hat{\mathbf{k}}_s^\top,$$

where

$$\mu_1 = \lambda_1 + \sum_{s=r+1}^{t} \lambda_s > 0.$$

Equivalently,

$$\mathbf{M}_t = \mathbf{A} \mathbf{\Lambda}' \mathbf{B}^\top,$$

where

$$\mathbf{A} = [\hat{\mathbf{v}}_1, \ldots, \hat{\mathbf{v}}_r] \in \mathbb{R}^{d \times r}, \qquad \mathbf{B} = [\hat{\mathbf{k}}_1, \ldots, \hat{\mathbf{k}}_r] \in \mathbb{R}^{m \times r},$$

and

$$\mathbf{\Lambda}' = \mathrm{diag}(\mu_1, \lambda_2, \ldots, \lambda_r).$$

The matrices $\mathbf{A}$ and $\mathbf{B}$ have full column rank $r$, and $\mathbf{\Lambda}'$ is invertible. Hence

$$\mathrm{rank}(\mathbf{M}_t) = \mathrm{rank}(\mathbf{A} \mathbf{\Lambda}' \mathbf{B}^\top) = r = \min(t, d, m).$$

Therefore, the upper bound is attainable.

**5. Generic Tightness.**   It remains to show that the rank-deficient case is non-generic. For fixed positive coefficients $\lambda_1, \ldots, \lambda_t$, the entries of

$$\mathbf{M}_t = \sum_{s=1}^{t} \lambda_s \hat{\mathbf{v}}_s \hat{\mathbf{k}}_s^\top$$

are polynomial functions of the direction coordinates $\{(\hat{\mathbf{v}}_s)_i, (\hat{\mathbf{k}}_s)_j\}_{s,i,j}$. The condition

$$\mathrm{rank}(\mathbf{M}_t) < r$$

is equivalent to the simultaneous vanishing of all $r \times r$ minors of $\mathbf{M}_t$. Each such minor is a polynomial in the direction coordinates.

The construction in Paragraph 4 shows that at least one $r \times r$ minor is not identically zero. Hence the set of direction pairs for which all $r \times r$ minors vanish is a proper algebraic subset of

$$(S^{d-1} \times S^{m-1})^t.$$

Such a proper algebraic subset has measure zero with respect to the product surface measure on the direction space. Therefore, under non-degenerate updates, the set of direction pairs $\{(\mathbf{v}_s, \mathbf{k}_s)\}_{s=1}^{t}$ for which

$$\mathrm{rank}(\mathbf{M}_t) < \min(t, d, m)$$

has measure zero. This proves that the upper bound is generically tight. □

## Appendix B: Training Details

In this appendix, we provide the parallel training algorithm for MuonSSM, followed by comprehensive details regarding the model architectures, training hyperparameters, and optimization settings used in our experiments across language, vision, and time-series modalities. All experiments were conducted on a single NVIDIA H100 GPU.

---

**Algorithm 1** MuonSSM: Parallel Mode (Training)

---

**Input:** $\mathbf{K} \in \mathbb{R}^{L \times m}, \mathbf{V} \in \mathbb{R}^{L \times d}, \mathbf{Q} \in \mathbb{R}^{L \times m}$
**Gates:** $\boldsymbol{\alpha}, \boldsymbol{\beta} \in \mathbb{R}^L$
**Params:** $\delta, \gamma, \tau, \eta$
// 1. Parallel Pre-computation
**for** $t = 1$ **to** $L$ **in parallel do**
  $\mathbf{X}_t \leftarrow \tau \beta_t \mathbf{v}_t \mathbf{k}_t^\top$
  $\mathbf{U}_t \leftarrow \text{NS}(\mathbf{X}_t)$
  $\mathbf{D}_t \leftarrow \alpha_t (\mathbf{I}_m - \beta_t \eta \mathbf{k}_t \mathbf{k}_t^\top)$
**end for**
// 2. Construct Associative Operators
**for** $t = 1$ **to** $L$ **in parallel do**
  $\boldsymbol{\Phi}_t \leftarrow \begin{bmatrix} \mathbf{D}_t & \mathbf{0}_{m \times m} \\ \gamma \mathbf{I}_m & \gamma \mathbf{I}_m \end{bmatrix} \in \mathbb{R}^{2m \times 2m}$
  $\boldsymbol{\Psi}_t \leftarrow \begin{bmatrix} \mathbf{U}_t & \mathbf{U}_t \end{bmatrix} \in \mathbb{R}^{d \times 2m}$
**end for**
// 3. Parallel Associative Scan
$\{\mathcal{Z}_t\}_{t=1}^L \leftarrow \text{Scan}(\{(\boldsymbol{\Phi}_t, \boldsymbol{\Psi}_t)\}_{t=1}^L, \oplus)^2$
// 4. Extract Memory States
$\mathbf{S}_{1:L} \leftarrow [\mathcal{Z}_1[:, : m], \dots, \mathcal{Z}_L[:, : m]]$
// 5. Compute Outputs
**for** $t = 1$ **to** $L$ **in parallel do**
  $\mathbf{y}_t \leftarrow \mathbf{S}_t \mathbf{q}_t$
**end for**
**Return** $\{\mathbf{y}_t\}_{t=1}^L$

---

## B.1. Parallel Training Algorithm

Algorithm 1 provides the parallel training procedure for MuonSSM. The key observation is that the coupled memory–momentum dynamics can be written as a block-affine recurrence, allowing the sequence to be evaluated using an associative scan. The Newton–Schulz normalization is applied locally at each timestep and therefore does not affect the asymptotic scan complexity.

## B.2. Language Modeling

We base our language modeling experiments on the Gated DeltaNet architecture, replacing the standard Delta layers with our proposed MuonSSM blocks. The model configuration follows a standard small-scale setting (170M parameters) suitable for rigorous ablation. The specific architectural parameters are detailed in Table 8.

Models are pre-trained on the FineWeb-Edu 10B token dataset using a causal language modeling objective. We utilize the AdamW optimizer with a cosine learning rate decay schedule. To ensure stability, we employ gradient clipping and a warmup period. The full training configuration is provided in Table 9.

## B.3. Vision Spatial Modeling

For visual representation learning, we adopt the training and evaluation protocols from MambaVision. We evaluate MuonSSM on three downstream tasks: Image Classification (ImageNet-1K), Object Detection (COCO), and Semantic Segmentation (ADE20K). We use the Tiny variant of the hierarchical architecture, replacing the spatial SSM mixers with MuonSSM layers. For object detection, we utilize the Mask R-CNN framework. For semantic segmentation, we employ the UperNet framework. Table 10 summarizes the specific hyperparameters used for each task.

## B.4. Time-Series: Human Activity Recognition

We evaluate MuonSSM on the Multi-Modal Activity (MMA) benchmarks. The experimental setup strictly controls for data preprocessing and model size to isolate algorithmic improvements.

**Data Preprocessing.** Raw inertial signals (tri-axial accelerometer and gyroscope) are resampled and segmented into fixed-length windows of $L = 512$ with a 50% overlap. Each channel is standardized to zero mean and unit variance using training set statistics. The resulting segments are formatted as tensors $X \in \mathbb{R}^{B \times L \times 6}$, where $B$ denotes the batch size.

*Table 8.* Language model architecture configuration based on Gated DeltaNet specifications.

| Hyperparameter | Value |
|---|---|
| Layers ($N$) | 10 |
| Heads ($H$) | 12 |
| Model Dimension ($D$) | 672 |
| Intermediate Dimension | 6144 |
| Context Length | 4096 |
| Vocab Size | 32000 |
| Local Window Size | 2048 |
| MuonSSM per Layer | 1 |
| Rotary Percentage | 1.0 (100%) |
| Normalization | FusedRMSNorm ($\epsilon = 1e-5$) |
| MLP Type | LLaMA MLP |
| Precision | bf16-mixed |

*Table 9.* Language model training hyperparameters for pre-training on FineWeb-Edu 10B.

| Hyperparameter | Value |
|---|---|
| **Data & Batching** | |
| Total Tokens | 10B ($1 \times 10^{10}$) |
| Global Batch Size | 512 sequences |
| Sequence Length | 4096 |
| **Optimization (AdamW)** | |
| Peak Learning Rate | $1 \times 10^{-3}$ |
| Min Learning Rate | $1 \times 10^{-4}$ |
| Weight Decay | 0.1 |
| Betas ($\beta_1, \beta_2$) | (0.9, 0.95) |
| Gradient Clipping | 1.0 |
| Warmup Tokens | 100M (1% of total) |

*Table 10.* Vision Training Hyperparameters. Settings for Classification, Object Detection, and Semantic Segmentation.

| Parameter | Classification | Object Detection | Semantic Seg. |
|---|---|---|---|
| Dataset | ImageNet-1K | MS COCO | ADE20K |
| Framework | - | Mask R-CNN | UperNet |
| Optimizer | LAMB | AdamW | AdamW |
| Learning Rate (LR) | 5e-3 / 1e-4 | 1e-4 | 5e-5 |
| Weight Decay | 0.05 | 0.05 | 0.01 |
| Stochastic Depth | 0.2 | 0.2 | 0.3 |
| Batch Size | 256 | 8 | 8 |
| Training Duration | 310 epochs | 36 epochs | 160K iters |

**Model Configuration.** The backbone consists of a lightweight Conv1D front-end (kernel size 3, stride 1) for local feature extraction, followed by $N = 2$ stacked MuonSSM layers with a hidden dimension $d_{model} = 128$. A global average pooling layer and a compact linear classification head produce the final predictions. Dropout with $p = 0.1$ is applied after the encoder and before the classification head.

**Training Setup.** All models are trained end-to-end using categorical cross-entropy loss. We use the Adam optimizer with the following schedule: Initial Learning Rate: $1 \times 10^{-3}$, Weight Decay: $1 \times 10^{-4}$, Scheduler: Cosine annealing without restarts, Batch Size 16, Epochs: 50 with Early Stopping, patience = 10, Gradient Clipping: Global norm set to 1.0 to stabilize training dynamics.

### B.5. Sensitivity Analysis and Robustness

To evaluate the robustness of MuonSSM and its sensitivity to hyperparameter choices, we performed a comprehensive post-hoc analysis over two key parameters: the momentum coefficient $\gamma \in \{0.0, 0.5, 0.8, 0.9, 0.95, 0.99\}$ and the input scaling factor $\tau \in \{0.4, 0.5, 0.6, 0.8, 1.0, 1.2\}$.

**Model Selection Protocol.** We emphasize that primary model selection was conducted strictly using validation sets. The results presented in Figure 6 serve as a post-hoc sensitivity analysis to demonstrate the stability of MuonSSM across diverse data distributions.

**Results and Discussion.** As illustrated in the heatmaps, MuonSSM exhibits consistent performance gains over the baseline across a wide, principled range of hyperparameters. Specifically, within the recommended range of $\gamma \in [0.8, 0.99]$ and $\tau \in [0.6, 0.8, 1.0]$ (12 configurations), the standard deviation in test accuracy remains minimal (e.g., $\pm 0.27\%$ for MuWiGes, $\pm 0.63\%$ for UESTC-MMEA-CL, and $\pm 0.72\%$ for MMAct).

Table 11 provides a quantitative summary of these 12 configurations compared to the standard LongHorn baseline:

This stability is theoretically grounded in our use of Newton–Schulz iteration, which bounds the singular values of updates ($\sigma_{max} \leq 1$), preventing uncontrolled amplification regardless of the specific scaling factor $\tau$. These results suggest that MuonSSM can be reliably deployed with a default configuration (e.g., $\gamma = 0.9, \tau = 0.6$) without the need for extensive per-dataset tuning.

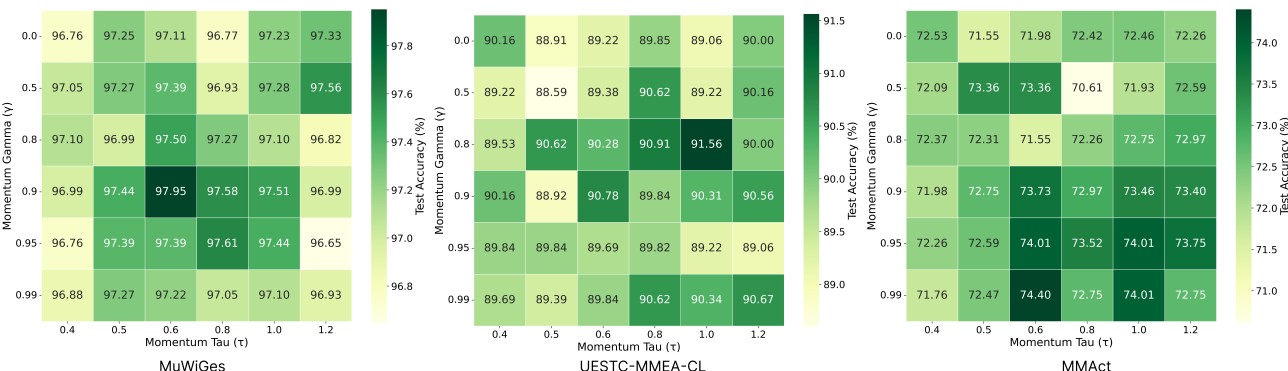

*Figure 6.* Post-hoc sensitivity analysis: Heatmap showing test accuracy (%) across different combinations of momentum tau ($\tau$) and momentum gamma ($\gamma$) parameters.

| Dataset | Baseline | Min (in range) | Max (in range) | % configs $\geq$ Base |
|---|---|---|---|---|
| MuWiGes | 97.23 | 97.05 | 97.95 | 67% (8/12) |
| UESTC-MMEA-CL | 89.06 | 89.22 | 91.56 | 100% (12/12) |
| MMAct | 72.47 | 71.55 | 74.40 | 83% (10/12) |

*Table 11.* Quantitative robustness summary across the principled hyperparameter range.

## Appendix C: Additional Empirical Analyses

### C.1. Empirical Evidence for NS-Induced Rank Enrichment

Remark 3.7 suggests that the backward geometry of Newton–Schulz (NS) normalization biases the learning signal toward directions less aligned with the current rank-one write. When accumulated through the momentum recurrence, these less collinear writes are expected to increase the effective rank of the momentum state $\mathbf{M}_t$. We empirically examine this mechanism using two complementary diagnostics.

**1. Three-way normalization ablation.** We compare three variants of the same backbone on MMAct under the same training setup: momentum alone, momentum with Frobenius normalization, and momentum with NS normalization. For each variant, we measure validation accuracy and the effective rank

$$r_{\text{eff}}(\mathbf{M}_t) = \frac{\left( \sum_i \sigma_i(\mathbf{M}_t) \right)^2}{\sum_i \sigma_i^2(\mathbf{M}_t)}. \tag{24}$$

A higher effective rank indicates that the singular values of $\mathbf{M}_t$ are more evenly distributed, and hence that the momentum state uses more independent representational directions.

Frobenius normalization provides only a small improvement over momentum alone, increasing the effective rank from 12.98 to 13.34. In contrast, NS increases the effective rank to 16.62 and improves validation accuracy by 2.14 percentage points over Frobenius normalization. This suggests that NS contributes more than magnitude normalization: its backward geometry encourages less redundant update directions when accumulated in $\mathbf{M}_t$.

**2. Rank truncation intervention.** We further test whether the additional rank is functionally useful. After training a deep MuonSSM model, we apply SVD to the internal SSM representations and retain only the top-$k$ singular components at inference time, while keeping all model parameters fixed. The results show a clear accuracy drop in the low-rank regime, indicating that the additional singular directions preserved by MuonSSM contribute to downstream prediction.

**Interpretation.** Together, these diagnostics support the mechanism described in Remark 3.7. Momentum accumulation can increase the nominal rank by summing rank-one writes, but repeated writes may still remain highly collinear. Frobenius normalization controls the scale of each write, whereas NS changes the backward geometry by emphasizing directions orthogonal to the current write. Empirically, this leads to higher effective rank and better downstream accuracy, consistent with the proposed rank-enrichment mechanism.

*Table 12.* Three-way ablation isolating the role of NS in rank enrichment on MMAct. NS yields both higher effective rank and better validation accuracy than momentum alone and momentum with Frobenius normalization.

| Variant | Val. Acc. ↑ | Effective Rank ↑ |
|---|---|---|
| Momentum only | $72.04 \pm 0.58$ | $12.98 \pm 0.81$ |
| Momentum + Frobenius | $72.53 \pm 0.82$ | $13.34 \pm 0.43$ |
| Momentum + Newton–Schulz | $\mathbf{74.67 \pm 0.46}$ | $\mathbf{16.62 \pm 0.57}$ |

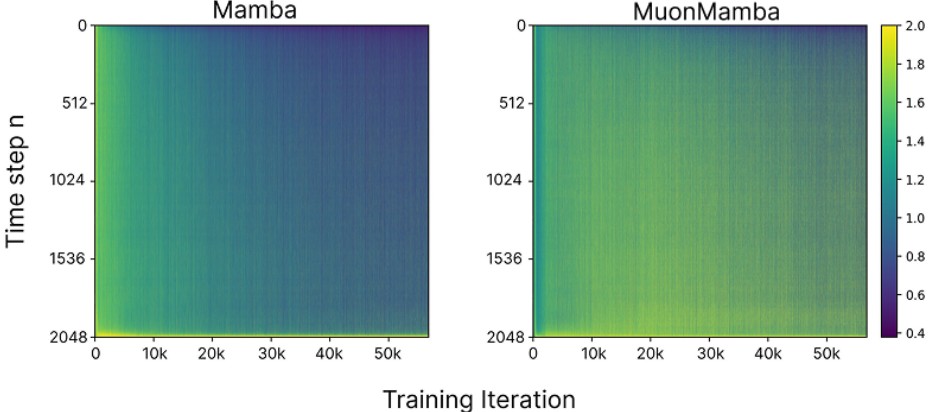

*Figure 7.* Gradient norm heatmaps over timesteps and training iterations. Each value is $\|\partial\mathcal{L}/\partial\mathbf{h}_n\|_2$. Compared with Mamba, MuonMamba shows more uniform gradient propagation across long contexts, supporting the structural gradient-preservation mechanism in Proposition 3.3.

## C.2. Gradient Propagation Heatmaps

Proposition 3.3 shows that the momentum pathway provides an additional route for gradient propagation through the $(\gamma\mathbf{I}_m)^{T-t+1}$ block. Since practical models include nonlinearities, gating, normalization, and deep stacking, we complement this linear analysis with an empirical gradient-propagation diagnostic.

We compare Mamba and MuonMamba under the same language modeling setup with sequence length 2048 over approximately 60K training iterations. At selected iterations, we record the gradient norm with respect to the hidden state at each timestep:

$$g(n) = \left\|\frac{\partial\mathcal{L}}{\partial\mathbf{h}_n}\right\|_2.$$

Figure 7 visualizes these values as heatmaps over timesteps and training iterations.

Vanilla Mamba exhibits a clear gradient decay pattern, with much smaller gradient norms for earlier tokens. In contrast, MuonMamba maintains a more uniform gradient profile across the sequence. This supports the interpretation that the momentum pathway mitigates long-range gradient attenuation in realistic nonlinear training, although it does not constitute a strict non-vanishing gradient guarantee.

