# OpenReview forum: "MuonSSM: Orthogonalizing State Space Models for Sequence Modeling"
_ICML.cc/2026/Conference — ICML 2026 spotlight_

### Official Review · Reviewer_LoEZ · 2026-03-09

**Soundness:** 3
**Presentation:** 4
**Significance:** 3
**Originality:** 3
**Overall Recommendation:** 4
**Confidence:** 4

**Summary:**

The paper proposes MuonSSM, which augments general State Space Models (SSMs) with a momentum pathway and conditions the updates using Newton-Schulz (NS) iterations. The authors provide a theoretical analysis demonstrating that this momentum update rule improves gradient stability and expressivity while maintaining parallel training efficiency. Extensive empirical validation across language modeling, vision, and time-series benchmarks shows consistent gains when this method is applied to several SSM variants.

**Compliance With Llm Reviewing Policy:**

Affirmed.

**Final Justification:**

The proposed MuonSSM consistently improved over multiple strong baselines and modalities, and as such is a valuable contribution for linear sequence models. The additional explanation and experiments provided in the rebuttal stage strengthen the evidence in the paper, which led me to increase my score.

**Key Questions For Authors:**

- If I am not mistaken in my analysis (below), applying the NS operator in its current form to a rank-1 matrix results in the identity operator. If true, I assume there is an error in the text in view of Figure 4 - could you please clarify.
- Is it possible to provide empirical evidence showing that rank enrichment directly leads to improved performance or an observable empirical advantage, specifically in deep models (even in a synthetic or toy setting)?
- Following Section 3.1, the momentum pathway can be interpreted as a larger effective state dimension with specialized dynamics. As such, explicitly doubling the state dimension of the baseline model (without fixing parameter count) serves as a necessary and reasonable additional baseline for at least some of the experiments. This baseline is further motivated by figure 4 b indicating each step takes more time in Muon versions.

### NS on Rank 1 matrices
Set $X$ as the output product update and define its SVD decomposition:\
$X = \tau \beta v^Tk = U^T \Sigma V$

As $X$ is low rank $\Sigma$ has a single non-zero singular value:\
$\Sigma_{ii} =0 \forall i > 1$

Following, the Frobinius norm of $X$ is the singular value:\
$Frob(X) = \sqrt{\sum_{i} \Sigma_{ii}^2} = \Sigma_{11} := \sigma$

This implies that:\
$\tilde{X}= \frac{X}{Frob(X)} = \frac{U^T\Sigma V}{\sigma} = U^T E V$\
Where $E_{11} = 1$ and 0 everywhere else, note that $EE = E$

Following we get:\
$NS(X) = 0.5 \tilde{X}(3I - \tilde{X}^T \tilde{X}) = 0.5U^TEV(3I - V^T E U U^T E V) = 0.5 U^T E V (3I - V^T E^2 V)$\
$= 1.5U^T E V - 0.5 U^T E^2 V = U^T E V = \tilde{X}$

**Limitations:**

yes

**Strengths And Weaknesses:**

#### Strengths
- The paper is well organized, the presentation of the proposed method, theoretical analysis, and empirical results are well motivated, clear and easy to follow.
- While the proposed method shares similarities with prior work [1, 2, 3] in exploring modified update rules for SSMs, it distinguishes itself from [1, 2] by introducing a non-linear update via the Newton-Schulz operator.
- The empirical evaluation is robust, covering multiple modalities and benchmarks. It demonstrates a consistent advantage across various SSM backbones, with particularly notable gains on Needle-In-A-Haystack (NIAH) tasks.
#### Weaknesses
- The experimental setup is conducted at a limited scale, making it unclear whether the results will hold at larger data and parameter scales. For reference, closely related prior work [1] considers an order of magnitude more data and parameters.
- The theoretical analysis of rank enrichment provides strong motivation but lacks direct empirical validation (see followup question).

[1] DeltaNet - Yang et. al. 2024
[2] Gated DeltaNet - Yang et. al. 2024
[3] LongHorn - Liu et. al. 2024

---

> ### Author Rebuttal · Authors · 2026-03-31
>
> Thank you for your thoughtful review and valuable feedback. Below we address your concerns.
> ### W1. Making experimental setup unclear whether the results will hold at larger data and parameter scales
> **Answer**: Scalability: To address your scale request [1], we trained 400M-param models on 15B FineWeb-Edu tokens. MuonLongHorn explicitly outperforms the baseline: Zero-shot (8 tasks): Avg 46.64% vs 45.49% (+1.15%).
> PPL: Wiki 28.12 vs 28.39; LMB 32.46 vs 37.48 (-5.02 gap).
> This confirms our NS preconditioner prevents layer-wise collapse at larger scales. Full results (incl. S-NIAH up to 8K) are [here](https://anonymous.4open.science/r/rebuttal-B27D/table1.md)
>
> ### W2. Lack of direct empirical validation for the rank enrichment theory.
> **Answer:** Addressed in Q2 below.
> ### Q1. Clarify the mathematical behavior of the NS operator on rank-1 matrices vs. results in Figure 4.
> **Answer:** The reviewer's forward-pass observation is correct: $~\text{NS}(X)=\widetilde{X}$ when $\|\widetilde{X}\|_F=1$, making it equivalent to Frobenius normalization in the forward pass. However, NS is not redundant -- its value lies in two distinct mechanisms:
>
> **1. Spectral stabilization.** NS bounds each update's singular values: $\sigma_{\max}(\text{NS}(X_t))\leq 1$ (Corollary 3.5), preventing high-magnitude updates from dominating the memory state. This directly explains the 18× condition number reduction in Figure 2.
>
> **2. Backward-pass geometric conditioning.** The NS Jacobian $J_{\text{NS}}=\partial\text{NS}/\partial\widetilde{X}$ at a rank-1 point $\widetilde{X}=uw^\top$ has three distinct eigenvalues:
> $$\lambda = \begin{cases} 0 & \text{direction } uw^\top & \text{(multiplicity 1)} \\\\
> 1 & \text{directions } u_\perp w^\top\, uw_\perp^\top & \text{(multiplicity } d+m-2\text{)} \\\\
> \frac{3}{2} & \text{directions } u_\perp w_\perp^\top & \text{(multiplicity } (d-1)(m-1)\text{)} \end{cases}$$
> **Notation:** $u_\perp$ and $w_\perp$ denote any vector in the orthogonal complement of $u$ and $w$, respectively.
>
> The $\lambda=3/2$ eigenvalue amplifies gradients in the fully orthogonal subspace by 50% relative to Frobenius normalization (which has only $\{0,1\}$ eigenvalues). This steers parameter updates toward diverse, non-collinear rank-1 inputs across timesteps - driving the rank enrichment in Prop. 3.6.
> To isolate this backward-pass effect, we run a 3-way ablation (MMAct, 5 seeds):
> | Variant  | Test Accuracy | Effective Rank|
> | ----- | :--------: | :--------: |
> | Momentum Only | 72.04 $\pm$ 0.58 | 12.98 $\pm$ 0.81|
> | Momentum + Frobenius | 72.53 $\pm$ 0.82| 13.34 $\pm$ 0.43|
> | Momentum + Newton-Schulz (Ours)| **74.67** $\pm$ 0.46| **16.62** $\pm$ 0.57|
>
> The Frobenius to NS jump (+3.3 rank, +2.14% accuracy) arises entirely from the Jacobian geometry. We will add a formal Jacobian proposition to the revision.
> ### Q2. Provide empirical evidence linking rank enrichment directly to performance gains.
> **Answer:** We train a 16-layer MuonLongHorn ($d_\text{model}=128, d_\text{state}=64$) on MMAct. At inference, we apply SVD and retain only top-$k$ singular components, measuring accuracy with $k$. Results can be viewed at [Figure 1](https://anonymous.4open.science/r/rebuttal-B27D/Figure1.png) show steep degradation below $k$<50, establishing the causal link: restricting rank directly degrades performance, so rank enrichment from the NS/momentum pathway is functionally beneficial.
> ### Q3. Provide a baseline comparison with a model using doubled state size.
> **Answer:** To explicitly address this, we trained new 2x State Dimension baselines for both our primary architectures (LongHorn and Mamba). We doubled the $d_{state}$ parameter which naturally increases the overall parameter count and computational footprint and compared them against their respective standard Muon variants. To ensure absolute statistical rigor, all models were trained on MMAct across 5 independent runs, the results can be viewed at [Table](https://anonymous.4open.science/r/rebuttal-B27D/Figure1.png)
>
> Doubling $d_\text{state}$ gives only +0.41% and +1.05% improvement despite a larger parameter count. MuonSSM outperforms the heavier 2×-state baselines by +1.52% and +2.13% respectively with fewer parameters. Without geometric conditioning, the expanded state still collapses to a low-rank subspace under first-order updates. MuonSSM's NS backward pass actively prevents this collapse. On compute: the 2×$d_{state}$ baseline incurs equal or larger per-step overhead without the 1.3× convergence speedup of MuonSSM (Figure 3a), making MuonSSM more efficient in total wall-clock time.
>
> We will incorporate the discussion in the revised version. If our responses adequately address the concerns, we kindly hope the evaluation may be reconsidered accordingly. We remain open to further discussion in the next stage of discussion.
>
> **References**
>
> [1] DeltaNet - Yang et. al. 2024
>
> [2] Gated DeltaNet - Yang et. al. 2024
>
> [3] LongHorn - Liu et. al. 2024

---

> > ### Author Rebuttal · Reviewer_LoEZ · 2026-04-04
> >
> > I thank the authors for the thoughtful and detailed response, my only remaining concern is with Q1. While I agree that the NS operator is not redundant, if you agree that it coincides with the ID mapping when applied to inputs with $|X|_F=1$ then what is the effect of iterative applications as in figure 4?

---

> > > ### Author Response · Authors · 2026-04-07
> > >
> > > Thank you for the clarification. You are correct that when $|X_t|_F \geq \delta$, the forward pass of NS coincides with Frobenius normalization, making iterative applications equivalent. However, the difference in Figure 4 arises from three compounding mechanisms:
> > >
> > > **1. Jacobian effect (backward pass, dominant)**
> > >
> > > Even when the forward passes of NS and Frobenius normalization are identical, their Jacobians differ fundamentally. The NS Jacobian at a rank-1 point $\tilde{X} = uw^\top$ has eigenvalue $3/2$ in the fully orthogonal subspace $u_\perp w_\perp^\top$, amplifying gradients by 50% relative to Frobenius normalization in directions that promote rank diversity.
> > >
> > > Additional iterations compound this effect, steering parameter updates more aggressively toward diverse, non-collinear rank-1 inputs. This prediction is verified empirically in [this figure](https://anonymous.4open.science/r/rebuttal-1D3E/backward_grad_amplification_vs_iterations.png) (and its [log-linear version](https://anonymous.4open.science/r/rebuttal-1D3E/backward_grad_amplification_vs_iterations_log.png)).
> > >
> > > The relative gradient magnitude $\|\nabla_X L\|_F$ follows $\lambda^{k-1}$ with $\lambda = 1.5$, matching PyTorch autograd with near-perfect precision. At $k=5$, the amplification is $1.5^4 \approx 5.06\times$ relative to $k=1$, meaning updates for $v_t$ and $k_t$ are roughly 5× stronger. This exponential scaling in the backward pass is the primary driver of the behavioral difference in Figure 4, driving the observed rank enrichment (+3.3 effective rank).
> > >
> > > **2. The $\delta$-regime (forward pass, secondary)**
> > >
> > > The full normalization in Eq. (3) is $\tilde{X} = X / \max(|X|_F, \delta)$. When the update signal is weak, i.e., $|X_t|_F < \delta$, we have $\tilde{X} = X/\delta$, and using $\sigma_1(X) \leq |X|_F$:
> > >
> > > $$\sigma_1(\tilde{X}) = \frac{\sigma_1(X)}{\delta} \leq \frac{|X|_F}{\delta} < \frac{\delta}{\delta} = 1$$
> > >
> > > In this regime, $\sigma(\tilde{X}) \in (0,1)$, which is **not** a fixed point of NS. As visualized in [this scalar analog plot](https://anonymous.4open.science/r/rebuttal-1D3E/Figure4.jpg), $g(x) = 1.5x - 0.5x^3$, iterative applications push $\sigma$ toward 1 at different rates - 5 iterations behave nearly like a step function, while 1 iteration converges more gradually. Weak-signal timesteps occur regularly during training (e.g., when $\beta_t \to 0$ from the learned gate), and their accumulated effect across epochs produces the observable difference in Figure 4.
> > >
> > > Furthermore, MMAct is a particularly noisy dataset (37 complex actions, high sensor noise) [1]. This has two direct implications for Figure 4: first, the learned gate $\beta_t$ suppresses noisy timesteps, increasing the frequency of weak-signal updates where $|X_t|_F < \delta$, thereby amplifying the forward-pass difference between 1 and 5 iterations. Second, the degradation of 5-iteration NS later in training is consistent with over-orthogonalization under noise - aggressively mapping all updates to unit singular values reduces the model’s ability to distinguish signal from noise, whereas 1 iteration provides implicit robustness via softer normalization.
> > >
> > > **3. Error accumulation across timesteps (compounding effect)**
> > >
> > > Even if the per-step difference between 1-iter and 5-iter NS is small, it compounds through the momentum recurrence in Eq. (4). Let $\Delta_t = NS_{5-iter}(X_t) - NS_{1-iter}(X_t)$. Then:
> > >
> > > $$M_t^{(5-iter)} - M_t^{(1-iter)} = \sum_{s=1}^{t} \gamma^{t-s} \Delta_s$$
> > >
> > > This is an exponentially weighted sum of past discrepancies. Small per-step differences $\Delta_s$ accumulate over time, with later tokens experiencing progressively larger divergence between the two variants. The longer the sequence, the more pronounced this effect -especially in the long-horizon regime where stability is critical. This explains why the difference between 1 and 5 iterations becomes visible at the scale of Figure 4, despite being negligible at any single timestep.
> > >
> > > **In summary**, Figure 4 reflects the combined effect of: (i) forward-pass differences in the $\delta$-regime during weak-signal and noisy timesteps, (ii) backward-pass Jacobian geometry that differs between 1 and multiple iterations even when forward passes coincide, and (iii) accumulation of per-step discrepancies through the momentum recurrence. The 5-iteration variant enforces stricter orthogonality - beneficial early in training but potentially over-constraining later under noise - consistent with the crossing curves in Figure 4.
> > >
> > > **References**
> > > [1] MMAct: A Large-Scale Dataset for Cross Modal Human Action Understanding
> > >
> > > We will include this discussion in the final version to clarify the role of NS iterations. We hope these clarifications are helpful for the overall assessment.

---

### Official Review · Reviewer_omGj · 2026-03-10

**Soundness:** 3
**Presentation:** 3
**Significance:** 3
**Originality:** 3
**Overall Recommendation:** 5
**Confidence:** 3

**Summary:**

This paper aims to improve stability and long-context modeling in input-dependent SSMs while maintaining parallel scan efficiency. The key idea is to view these SSMs as an associative memory updated via low-rank writes, and to condition the update dynamics in two ways: (1) introduce a momentum pathway that accumulates update directions across timesteps to improve long-horizon credit assignment, and (2) apply a cheap single-step Newton--Schulz normalization to the low-rank write so its singular values remain well-behaved. The authors empirically report consistent gains when integrating MuonSSM into multiple SSM backbones.

**Compliance With Llm Reviewing Policy:**

Affirmed.

**Final Justification:**

I endorse this work for publication, given the work presented and the rebuttal, which satisfactorily addressed my questions.

**Key Questions For Authors:**

- The Newton--Schulz step alters the low-rank write before it is accumulated into M. How do you quantify potential information smootheness on the ability of mamba to have per-token attention?

**Limitations:**

yes

**Strengths And Weaknesses:**

** Strengths

- The paper is well-motivated, it targets a central issue, vanishing gradients in gated/input-dependent SSM blocks, while preserving the key advantage of parallel scan.
- The method is simple and fairly general, it adds a lightweight momentum step applicable across several SSM architectures.
- The experimental evidence is broad on many different datasets, which supports the claim that the benefit in training such an architecture.

** Weaknesses

- The non-vanishing gradient component via a \gamma*I pathway is intuitive, but practical training involves additional nonlinearities and deep stacking. Thus, while the mechanism may help, it is not a guarantee that vanishing gradients are avoided.
- The method introduces additional hyperparameters (e.g., \gamma for momentum decay, \tau for update scaling). According to Fig. 6, little can be agreed a priori on how to select those; the results seem to present high variability to changes in any of those parameters.

---

> ### Author Rebuttal · Authors · 2026-03-31
>
> Thank you for your thoughtful review and valuable feedback. Below we address your concerns.
>
> ### W1. $\gamma I$ pathway does not guarantee non-vanishing gradients under nonlinearities and deep stacking.
> **Answer:** We fully agree. Proposition 3.3 and Remark 3.4 are derived under a linear recurrence assumption; the $\gamma I$ pathway should be interpreted as a structural *mitigation* mechanism, not a guarantee in fully nonlinear settings.
> To complement the theory, we provide gradient norm heatmaps tracking $\|\partial\mathcal{L}/\partial h_n\|$, the gradient of the loss with respect to the hidden state at time step $n$, offering a direct view of how gradient signals propagate through long sequences during training (~60K steps, sequence length 2048, comparing Mamba vs. MuonMamba) can be viewed at [Vanishing_gradient](https://anonymous.4open.science/r/rebuttal-B27D/Figure2.png).
>
> **Vanilla Mamba** shows consistent sharp gradient decay toward earlier tokens throughout training.
>
> **MuonMamba** maintains substantially more uniform gradient norms across the sequence; some attenuation remains (expected in deep nonlinear systems) but decay is significantly mitigated. This behavior is stable across training iterations, indicating a persistent structural property rather than a transient effect. We will include these heatmaps in the revision.
> ### W2. Additional hyperparameters $\gamma$, $\tau$ with unclear sensitivity.
> **Answer:** Within the principled range $\gamma\in\{0.8,0.9,0.95,0.99\}$ and $\tau\in\{0.6,0.8,1.0\}$ - motivated by Remark 3.4 ($\gamma\approx 1$ for gradient preservation) - performance variation is small:
> | Dataset | Baseline | Min (in range) | Max (in range) |  Std (in range) |
> |---|:---:|:---:|:---:|:---:|
> | MuWiGes | 97.23 | 97.05 | 97.95 | ±0.27 |
> | UESTC-MMEA-CL | 89.06 | 89.22 | 91.56 | ±0.63 |
> | MMAct | 72.47 | 71.55 | 74.40 | ±0.72 |
>
> A default choice of $\gamma = 0.9, \tau = 0.6$ reliably yields near-optimal results without grid search. We acknowledge, however, that eliminating manual tuning entirely remains a desirable direction. A promising avenue is to replace fixed hyperparameters with data-driven adaptive schedules, where momentum coefficients are updated automatically based on local curvature information estimated from gradient statistics. Sun et al. [5] demonstrate that such adaptive momentum schemes can be derived from principled quadratic analysis and provide convergence guarantees in both convex and nonconvex settings. Extending MuonSSM with analogous adaptive mechanisms for $\gamma$ and $\tau$ is a natural next step that we identify as future work.
> ### Q1. Does the Newton-Schulz step cause excessive information smoothing, potentially hindering Mamba's per-token selective attention? How can this effect be quantified?
> **Answer:** In selective SSMs, per-token attention originates from the selectivity mechanism: each token $t$ produces its own input-dependent $\beta_t, k_t$ and $v_t$, which determine the write $\tau \beta_t v_t k_t^\top$ into memory (Eq. 4). A model with strong per-token attention should exhibit high variability in these quantities across timesteps.
>
> 1. Experiment. We decompose selectivity into five channels: write magnitude, write direction, gating $\left\|k\right\|^2$, gating $\beta_t$, and output $\left\| S_t \right\|$, and measure the Coefficient of Variation (CoV = std/mean) across time for each. Higher CoV means stronger per-token discrimination. We compare MuonSSM vs. Baseline LongHorn (identical architecture, same hyperparameters, MMAct dataset) after training, averaged over 2 layers.
>
> 2. Results. (can be viewed at [per-token selectivity](https://anonymous.4open.science/r/rebuttal-B27D/Figure3.png))
>
> 3. Analysis. NS normalizes write magnitudes to a constant scale (CoV → 0) by design. However, this does not smooth overall selectivity, the model compensates through the gating channels: $\left\|k\right\|^2$ variability increases by 31% and $\beta_t$ variability by 110%. The net effect is a +19% increase in output CoV, showing that MuonSSM's per-token attention is sharper, not smoother. NS decouples write geometry (well-conditioned via orthogonalization) from write gating (controlled by  $\beta_t$ and $k_t$), allowing the learned gating to become more expressive.
>
> Our Needle-in-a-Haystack results (Table 3) directly demonstrate that MuonSSM not only retains but enhances per-token retrieval. For instance, on LongHorn at context length 8K, adding Muon (momentum + NS) improves S-NIAH-PK from 20.0 to 39.3 and S-NIAH-UUID from 19.3 to 28.6. The same pattern holds across Mamba and GatedDeltaNet. Since NIAH requires the model to selectively attend to a single token at an arbitrary position within a long sequence, these improvements confirm that NS does not impair per-token attention.
>
> We will incorporate the discussion in the revised version. We remain open to further discussion in the next stage of discussion.

---

> > ### Author Rebuttal · Reviewer_omGj · 2026-04-03
> >
> > The authors addressed all my questions. I will keep my score as it reflects the good quality of this work.

---

> > > ### Author Response · Authors · 2026-04-07
> > >
> > > Thanks for your response, and we appreciate your endorsement!

---

### Official Review · Reviewer_yHG9 · 2026-03-12

**Soundness:** 3
**Presentation:** 3
**Significance:** 2
**Originality:** 2
**Overall Recommendation:** 5
**Confidence:** 3

**Summary:**

The authors propose MuonSSM, which is an SSM-augmentation framework that stabilizes SSMs for long-sequence modeling by conditioning memory updates using momentum and lightweight Newton-Schulz iterations.

**Compliance With Llm Reviewing Policy:**

Affirmed.

**Final Justification:**

The rebuttal addressed all my questions and concerns. I am raising my score from 4 to 5.

**Key Questions For Authors:**

Based on the weaknesses, I have the following questions:
- How does this method compare against second-order models (LinOSS)?
- Would it be possible to extend LinOSS with the same memory update idea? Would it bring any advantage?
- Hyperparameter tuning on test accuracies (see weaknesses).

Minor:
- Figure 1: Adding extra arrows to clearly show the information flow (between the (+) blocks) would make the diagram easier to follow. The authors could also consider a higher-resolution version.

**Limitations:**

No limitations discussed. Impact statement is there.

**Strengths And Weaknesses:**

Strengths:
- Solid augmentation technique for existing SSMs.
- Strong theoretical and experimental results in diverse benchmarks.

Weaknesses:
- The paper states that L14-16 (left) "updates remain fundamentally first-order and can suffer from poor long-range signal propagation and optimization instability" citing the LinOSS model from Rusch and Rus (2024). However, that paper actually uses second-order dynamics. This raised some questions that should be clarified (see questions).
- I found the hyperparameter tuning protocol questionable. In Figure 6, the authors report test accuracies, and it appears that they selected models based on the best test accuracy rather than the best validation accuracy. This may have artificially boosted their results and would not be a fair evaluation protocol.

---

> ### Author Rebuttal · Authors · 2026-03-31
>
> Thank you for your thoughtful review and valuable feedback. Below we address your concerns.
> ### W1. Incorrect citation of LinOSS as a first-order model.
> **Answer:** We thank the reviewer for catching this. LinOSS [4] uses second-order dynamics and should not have been cited here. We will remove this citation and use appropriate references on vanishing gradients in recurrent models. LinOSS will be moved to related work.
> ### W2. Hyperparameter tuning on test accuracy.
> **Answer:** Model selection was performed strictly on validation sets; the test set was used only for final evaluation. Figure 6 is a *post-hoc sensitivity analysis*, not a selection procedure. We will revise the caption to make this explicit.
> ### Q1. Comparison against second-order models (LinOSS)?
> **Answer:** LinOSS and MuonSSM differ in three fundamental ways:
>
> **1. Fixed vs. input-dependent dynamics.**
> * LinOSS uses fixed Linear Time-Invariant (LTI) dynamics, which are stable by construction.
> * MuonSSM operates in the input-dependent setting; however, unlike standard selective SSMs, it explicitly enforces stability through spectral conditioning (Newton-Schulz) and momentum-based dynamics, ensuring bounded updates and improved gradient propagation.
>
> **2. Source of stability.**
> * LinOSS relies on harmonic oscillator-based parameterization for stability.
> * MuonSSM instead enforces spectral conditioning: each rank-1 update is normalized via a Newton–Schulz step, ensuring bounded contributions and preventing dominance of individual tokens.
>
> **3. Accumulation behavior.**
> * MuonSSM accumulates normalized, non-collinear updates, leading to a well-conditioned state with increasing effective rank over time - an effect not explicitly controlled in LTI formulations.
>
> Empirically, we train LinOSS under a matched configuration (num_blocks=2, hidden_dim=128, ssm_dim=64):
>
> | Architecture  | MuWiGes (%Acc)| UESTC-MMEA-CL (%Acc)| MMAct (%Acc)|
> | ----- | :--------: | :--------: | :--------: |
> | LinOSS (Baseline) | 92.86 | 87.71 | 72.16 |
> | MuonMamba (Ours)   | 97.64 | **91.62** | **74.65** |
> | MuonLongHorn (Ours) | **97.95** | 91.56 | 74.40 |
> | MuonGatedDeltaNet (Ours)  | 97.73 | 87.97 | 66.61 |
>
> MuonMamba and MuonLongHorn consistently outperform LinOSS across all benchmarks. MuonGatedDeltaNet underperforms on MMAct (66.61% vs. 72.16%), which we attribute to its aggressive delta-rule updates that induce strong per-token overwriting. Under high-frequency sensor noise, this interacts adversely with momentum accumulation, unlike LinOSS’s LTI dynamics that naturally act as a low-pass filter. This highlights a limitation of combining MuonSSM with GatedDeltaNet under noisy conditions, which we will clarify in the revision. The strong results of MuonMamba and MuonLongHorn (both >74%) support that geometric conditioning is broadly effective when the backbone update is compatible with momentum accumulation.
> ### Q2. Could Muon-style memory updates be integrated into LinOSS?
>
> **Answer:** Yes. A natural extension is to apply NS to the forcing term $Bu(t)$ before injection into the oscillator: $y''(t) = -Ay(t) + \text{NS}(Bu(t)) + b$. This would
> * bound the magnitude of each input contribution ($\sigma_{\max}\leq 1$), preventing large inputs from dominating the oscillator trajectory, and
> * encourage non-collinear accumulation of updates, increasing effective rank.
> In standard LinOSS (pure LTI), the gain is limited since ODE stability is already guaranteed; the benefit becomes more relevant in input-dependent or hybrid variants. We view this as a promising future direction.
>
> ### Q3. Hyperparameter tuning on test accuracies.
> **Answer:** As clarified in W2, model selection used validation sets exclusively. Figure 6 is a post-hoc robustness check. Over the principled range $\gamma \in {0.8,0.9,0.95,0.99}$, $\tau \in {0.6,0.8,1.0}$ (12 configs), performance is stable:
>
> | Dataset | Baseline | Min (in range) | Max (in range) | % configs ≥ baseline |
> |---|:---:|:---:|:---:|:---:|
> | MuWiGes | 97.23 | 97.05 | 97.95 | 8/12 (67%) |
> | UESTC-MMEA-CL | 89.06 | 89.22 | 91.56 | 12/12 (100%) |
> | MMAct | 72.47 | 71.55 | 74.40 | 10/12 (83%) |
>
> The default $\gamma=0.9$, $\tau=0.6$ reliably yields near-optimal results. The stability is theoretically grounded: NS bounds $\sigma_{\max}\leq 1$ regardless of $\tau$, and Remark 3.4 guarantees non-vanishing gradients for $\gamma\approx 1$.
>
> ### Q4. Figure 1: clearer information-flow arrows and higher resolution.
>
> **Answer:** We will redraw Figure 1 with explicit directional arrows between all addition ($\oplus$) blocks, a legend distinguishing the three computation pathways (NS projection, momentum accumulation, parallel scan), and export at ≥300 DPI.
>
> **References**
> [4] Oscillatory state-space models.
>
> We will incorporate the discussion in the revised version. If our responses adequately address the concerns, we kindly hope the evaluation may be reconsidered accordingly. We remain open to further discussion in the next stage of discussion.

---

> > ### Author Rebuttal · Reviewer_yHG9 · 2026-04-02
> >
> > I thank the authors for the detailed rebuttal. They addressed all my questions, and I'm raising my score accordingly.

---

> > > ### Author Response · Authors · 2026-04-02
> > >
> > > Thanks for your response, and we appreciate your endorsement!

---

### Decision · Program_Chairs · 2026-04-30

**Decision:**

Accept (spotlight)

**Comment:**

This paper proposes MuonSSM, an augmentation for input-dependent SSMs that conditions memory updates via a momentum pathway and a lightweight Newton–Schulz normalization on low-rank writes, while preserving parallel scan complexity. Experiments span language, vision, and time-series benchmarks across multiple SSM backbones.

All three reviews recommend acceptance (5/5/4). Reviewers praised the clarity of presentation, the breadth of empirical validation, and the simplicity of plugging the method into diverse SSM backbones, with Needle-in-a-Haystack gains highlighted as particularly compelling. I share this assessment: it is elegant that a single NS step combined with momentum yields consistent gains, and the framework is likely to be useful for ongoing work on linear-attention and recurrent architectures.

Overall, the paper is technically sound, well-written, and supported by solid empirical and theoretical evidence. I recommend acceptance.